# Planning with Theory of Mind for Few-Shot Adaptation in Sequential Social Dilemmas

## Abstract

Despite the recent successes of multi-agent reinforcement learning (MARL) algorithms, efficiently adapting to other agents in mixed-motive environments remains a significant challenge. One feasible approach is to use Theory of Mind (ToM) to reason about the mental states of other agents and model their behavior. However, these methods often encounter difficulties in efficient reasoning and utilization of inferred information. To address these issues, we propose Planning with Theory of Mind (PToM), a novel multi-agent algorithm that enables few-shot adaptation to unseen policies in sequential social dilemmas (SSDs). PToM is hierarchically composed of two modules: an opponent modeling module that utilizes ToM to infer others' goals and learn corresponding goal-conditioned policies, and a planning module that employs Monte Carlo Tree Search (MCTS) to identify the best response. Our approach improves efficiency by updating beliefs about others' goals both between and within episodes and by using information from the opponent modeling module to guide planning. Experimental results demonstrate that in three representative SSD paradigms, PToM converges expeditiously, excels in self-play scenarios, and exhibits superior few-shot adaptation capabilities when interacting with various unseen agents. Furthermore, the emergence of social intelligence during our experiments underscores the potential of our approach in complex multi-agent environments.

## 1 Introduction

Constructing agents being able to rapidly adapt to previously unseen agents is a longstanding challenge for Artificial Intelligence. We refer to this ability as few-shot adaptation. Previous work has proposed well-performed MARL algorithms to study few-shot adaptation in zero-sum games (Vinyals et al., 2019; Vezhnevets et al., 2020) and common-interest environments (Barrett et al., 2011; Hu et al., 2020; Mahajan et al., 2022; Team et al., 2023). These environments involve a predefined competitive or cooperative relationship between agents. However, little attention has been given to the challenge of adapting to new opponents[1] in mixed-motive environments, where cooperation coexists with defection. A majority of realistic multi-agent decision-making scenarios can be abstracted into mixed-motive environments (Komorita & Parks, 1995; Dafoe et al., 2020).

We focus on few-shot adaptation of unseen agents in sequential social dilemmas (SSDs), a widely-studied kind of mixed-motive environments. SSDs extend classic matrix-form social dilemmas temporally and spatially. They enable the observation of others' trajectories and modification of one's own strategies within one episode (Leibo et al., 2017). SSDs are inherently complex, requiring the dynamic identification of potential partners and competitors. Decision making in SSDs should balance short-term interests with long-term rewards, while also considering the trade-off between self-interest and group benefit. Many algorithms struggle to perform well in SSDs despite success in zero-sum and pure-cooperative environments, because they use efficient techniques specific to reward structures, such as minimax (Littman, 1994; Li et al., 2019), Double Oracle (McMahan et al., 2003; Balduzzi et al., 2019) or IGM condition (Sunehag et al., 2017; Son et al., 2019; Rashid et al., 2020), which are not applicable in SSDs. These challenges make autonomous decision-making and few-shot adaptation more difficult in SSDs compared with zero-sum and pure-cooperative environments.

---

[1]In this paper, we use "opponent" and "other agent" interchangeably to refer to agents that coexist with the focal agent in the same environment.

According to cognitive psychology and related disciplines, humans' ability to rapidly solve previously unseen problems depends on hierarchical cognitive mechanisms (Butz & Kutter, 2016; Kleiman-Weiner et al., 2016; Eppe et al., 2022). This hierarchical structure unifies high-level goal reasoning with low-level action planning. Meanwhile, researches on machine learning also emphasize the importance and effectiveness of hierarchical goal-directed planning for few-shot problem-solving (Eppe et al., 2022). Inspired by the hierarchical structure and theory of mind - the ability to understand others' mental states (like goals and beliefs) from their actions (Baker et al., 2017), we propose an algorithm, named Planning with Theory of Mind (PToM), for tackling few-shot adaptation in SSDs.

PToM consists of two modules: an opponent modeling module and a planning module. The opponent modeling module estimates opponents' behavior by inferring their goals and learning their goal-conditioned policies. Based on the opponent's behavior, the planning module generates the next action to take. To test PToM's few-shot adaptation ability, we construct three typical SSD environments: sequential stag-hunt game (SSH), sequential snowdrift game (SS), and sequential prisoner's dilemma (SPD). They are extensions of the three most representative paradigms of social dilemmas(Rousseau, 1999; Rapoport & Chammah, 1966; Rapoport et al., 1965; Santos et al., 2006), in terms of space, time, and number of participants. A detailed description of these environments is provided in Sec. 5.1.

Experimental results illustrate that across all the three typical paradigms of SSDs, PToM exhibits superior few-shot adaptation ability compared with baselines, including the well-established MARL algorithms LOLA, social influence, A3C, and prosocial-A3C. Meanwhile, PToM exhibits expeditious convergence and achieves high rewards after convergence, showing its exceptional decision-making ability in SSDs. In addition, we observe self-organized cooperation and alliance of the disadvantaged emerging from the interaction between multiple PToM agents.

## 2 RELATED WORK

MARL has explored multi-agent decision-making in SSDs. One approach is to add intrinsic rewards to incentivize collaboration and consideration of the impact on others, alongside maximizing extrinsic rewards. Notable examples include ToMAGA (Nguyen et al., 2020), MARL with inequity aversion (Hughes et al., 2018), and prosocial MARL (Peysakhovich & Lerer, 2018). However, many of these algorithms rely on hand-crafted intrinsic rewards and assume access to other agents' rewards, which can make them exploitable by self-interested algorithms and less effective in realistic scenarios where others' rewards are not visible (Komorita & Parks, 1995). To address these issues, Jaques et al. (2019) have included intrinsic social influence reward that use counterfactual reasoning to assess the effect of an agent's actions on its opponents' behavior.

LOLA Foerster et al. (2018) and its extension (such as POLA (Zhao et al., 2022), M-FOS (Lu et al., 2022)) consider the impact of one agent's learning process, rather than treating them as a static part of the environment. However, LOLA requires knowledge of opponents' network parameters, which may not be feasible in many scenarios. LOLA with opponent modeling relaxes this requirement, but scaling problems may arise in complex sequential environments that require long action sequences for rewards.

Our work relates to opponent modeling (see (Albrecht & Stone, 2018) for a comprehensive review). I-POMDP (Gmytrasiewicz & Doshi, 2005) is a typical opponent modeling and planning framework, which maintains dynamic beliefs over the physical environment and beliefs over other agents' beliefs. It maximizes a value function of the beliefs to determine the next action. However, the nested belief inference suffers from serious computational complexity problems, which makes it impractical in complex environments. Unlike I-POMDP and its approximation methods (Doshi & Perez, 2008; Doshi & Gmytrasiewicz, 2009; Hoang & Low, 2013; Han & Gmytrasiewicz, 2018; 2019; Zhang & Doshi, 2022), PToM explicitly uses beliefs over other agents' goals and policies to learn a neural network model of other agents (MOA), an MCTS planner to compute next actions. PToM avoids nested belief inference and performs sequential decision-making more efficiently.

Theory of mind (ToM), originally a concept of cognitive science and psychology (Baron-Cohen et al., 1985), has been transformed into computational models over the past decade and used to infer agents' mental states such as goals and desires. Bayesian inference has been a popular technique used to make ToM computational (Baker et al., 2011; Track et al., 2018; Wu et al., 2021; Zhi-Xuan et al., 2022). With the rapid development of the neural network, some recent work has attempted to achieve ToM using neural networks (Rabinowitz et al., 2018; Shu & Tian, 2018; Wen et al., 2019;

Moreno et al., 2021). PToM gives a practical and effective framework to utilize ToM, and extend its application scenarios to SSDs, where both competition and cooperation are involved and the goals of opponents are private and volatile.

Monte Carlo Tree Search (MCTS) is a widely adopted planning method for optimal decision-making. Recent work, such as AlphaZero (Silver et al., 2018) and MuZero (Schrittwieser et al., 2020) have used MCTS as a general policy improvement operator over the base policy learned by neural networks. However, MCTS is limited in multi-agent environments, where the joint action space grows rapidly with the number of agents (Choudhury et al., 2022). We avoid this problem by estimating opponent policies and planning only for the focal agent's actions.

## 3 PROBLEM FORMULATION

We consider multi-agent hierarchical decision-making in SSDs, which can be described as a Markov game (Liu et al., 2022) with goals, specified by a tuple $< N, S, \mathbf{A}, T, \mathbf{R}, \gamma, T_{max}, \mathbf{G} >$.

Here, agent $i \in N = \{1, 2, \cdots, n\}$ chooses action from action space $A_i = \{a_i\}$. $\mathbf{A} = A_1 \times A_2 \times \cdots \times A_n$ is the joint action space. The joint action $\boldsymbol{a}_{1:n} \in \mathbf{A}$ will lead to a state transition based on the transition function $T : S \times \mathbf{A} \times S \to [0, 1]$. Specifically, after agents take the joint action $\boldsymbol{a}_{1:n}$ the state of the environment will transit from $s$ to $s'$ with probability $T(s'|s, \boldsymbol{a}_{1:n})$. The reward function $R_i : S \times \mathbf{A} \to \mathbb{R}$ denotes the immediate reward received by agent $i$ after joint action $\boldsymbol{a}_{1:n}$ is taken on state $s \in S$. The discount factor for future rewards is denoted as $\gamma$. $T_{max}$ is the maximum length of an episode. $\pi_i : S \times A_i \to [0, 1]$ denotes agent $i$'s policy, specifying the probability $\pi_i(a_i|s)$ that agent $i$ chooses action $a_i$ at state $s$.

The environments we study have a set of goals, denoted by $\mathbf{G} = G_1 \times G_2 \times \cdots \times G_n$, where $G_i = \{g_i\}$ represents the set of goals for agent $i$. For any two agents $i$ and $j$, $j$'s true goal is inaccessible to $i$. However, $i$ can infer $j$'s goal based on its action sequence. Specifically, $i$ maintains a belief over $j$'s goals, $b_{ij} : G_j \to [0, 1]$, which is a probability distribution over $G_j$.

Here, algorithms are evaluated in terms of self-play and few-shot adaptation to unseen policies in SSDs. Self-play involves multiple agents using the same algorithm to undergo training from scratch. The performance of algorithms in self-play is evaluated by their expected reward after convergence. Self-play performance demonstrates the algorithm's ability to make autonomous decisions in complex and dynamic SSDs. Few-shot adaptation refers to the capability to recognize and respond appropriately to unknown policies within a limited number of episodes. The performance of algorithms in few-shot adaptation is measured by the rewards they achieve after engaging in these brief interactions.

## 4 METHODOLOGY

In this section, we propose **P**lanning with **T**heory **o**f **M**ind (PToM), a novel algorithm for multi-agent decision-making in SSDs. PToM consists of two main modules: an opponent modeling module to infer opponents' goals and predict their behavior and a planning module to plan the focal agent's best response guided by the inferred information from the opponent modeling module.

Based on the hypothesis in cognitive psychology that others' behavior is goal-directed (Gergely et al., 1995; Buresh & Woodward, 2007), and that agents behave stably for a specific goal (Warren, 2006), the opponent modeling module models opponent behavior with two levels of hierarchy. At the high-level, the module employs ToM to infer opponents' internal goals by analyzing their action sequences. Based on the inferred goals and the current state of the environment, the low-level component learns goal-conditioned policies to model the atomic actions of opponents.

In the planning module, MCTS is used to plan for the best response of the focal agent based on the inferred opponents' policies. To handle the uncertainty over opponents' goals, we sample multiple opponent goal combinations from the current belief and return the action that maximizes the average return over the sampled configurations. Following AlphaZero (Silver et al., 2018) and MuZero (Schrittwieser et al., 2020), we maintain a policy and a value network to boost MCTS planning and in turn use the planned action and its value to update the neural network.

Figure 1 gives an overview of PToM, and the pseudo-code of PToM is provided in Appendix A.

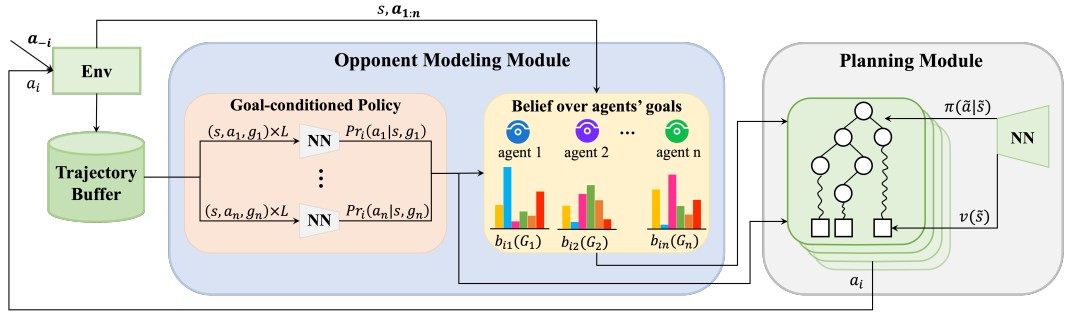

Figure 1: Overview of PToM. PToM consists of an opponent modeling module and a planning module. The opponent modeling module models opponent behavior by inferring opponents' goals and learning their goal-conditioned policies. Estimated opponent behavior is then fed to the planning module to select a rewarding action of the focal agent.

## 4.1 OPPONENT MODELING WITH EFFICIENT ADAPTATION

In goal-inference (as the light yellow component shown in Figure 1), PToM summarizes the opponents' objectives based on the interaction history. However, it faces the challenge of the opponent's goals potentially changing within episodes. To solve these issues, we propose two update procedures based on ToM: intra-ToM, which infers the opponent's immediate goals within a single episode, and inter-ToM, which summarizes the opponent's goals based on their historical episodes.

Intra-ToM reasons about the goal of opponent $j$ in the current episode $K$ according to $j$'s past trajectory in episode $K$. It ensures that PToM is able to quickly respond to in-episode behavior changes of other agents. Specifically, in episode $K$, agent $i$'s belief about agent $j$'s goals at time $t$, $b_{ij}^{K,t}(g_j)$, is updated according to:

$$
\begin{aligned}
b_{ij}^{K,t+1}(g_j) &= Pr(g_j \,|s^{K,0:t+1}, a_j^{K,0:t}) \\
&= \frac{Pr(g_j|s^{K,0:t}, a_j^{K,0:t-1}) Pr(a_j^{K,t}|s^{K,0:t}, a_j^{K,0:t-1}, g_j) Pr(s^{K,t+1}|s^{K,0:t}, a_j^{K,0:t}, g_j)}{Pr(s^{K,t+1}, a_j^{K,t}|s^{K,0:t}, a_j^{K,0:t-1})} \\
&= \frac{1}{Z_1} b_{ij}^{K,t}(g_j) Pr_i(a_j^{K,t}|s^{K,0:t}, g_j),
\end{aligned}
\tag{1}
$$

where $Z_1$ is the normalization factor that makes $\sum_{g_j \in G_j} b_{ij}^{K,t+1}(g_j) = 1$. The likelihood term $Pr_i(a_j^{K,t}|s^{K,0:t}, g_j)$ is provided by the goal-conditioned opponent policies, whose detailed description is given in the following.

However, intra-ToM may suffer from inaccuracy of the prior (i.e., $b_{ij}^{K,0}(g_j)$) when past trajectories are not long enough for updates. Inter-ToM makes up for this by calculating a precise prior based on past episodes. Belief update between two adjacent episodes is defined as:

$$
b_{ij}^{K,0}(g_j) = \frac{1}{Z_2}[\alpha b_{ij}^{K-1,0}(g_j) + (1-\alpha)\mathbf{1}(g_j^{K-1} = g_j)],
\tag{2}
$$

where $\alpha \in [0, 1]$ is the horizon weight, which controls the importance of the history. As $\alpha$ decreases, agents attach greater importance to recent episodes. $\mathbf{1}(\cdot)$ is the indicator function. $Z_2$ is the normalization factor. The equation is equivalent to a time-discounted modification of the Monte Carlo estimate. Inter-ToM summarizes other agents' goals according to all the previous episodes, which is of great help when playing with the same agents in a series of episodes.

The goal-conditioned policy (as the light yellow component shown in Figure 1) $\pi_{\boldsymbol{\omega}}(a_j^{K,t}|s^{K,0:t}, g_j)$, which is obtained through a neural network $\boldsymbol{\omega}$.

To train the network, a set of $(s_j^{K,t}, a_j^{K,t}, g_j^{K,t})$ is collected from episodes and sent to the replay buffer. $\boldsymbol{\omega}$ is updated at intervals to minimize the cross-entropy loss:

$$
L(\boldsymbol{\omega}) = \mathbb{E}[-\sum_{a \in A_j} \mathbf{1}(a_j^{K,t} = a) \log(\pi_{\boldsymbol{\omega}}(a|s^{K,0:t}, g_j^{K,t}))].
\tag{3}
$$

### 4.2 PLANNING UNDER UNCERTAIN OPPONENT MODELS

Given the policies of other agents estimated by the opponent modeling module, we can leverage planning algorithms such as MCTS to compute an advantageous action. However, a key obstacle to applying MCTS is that opponent policies estimated by the opponent modeling module contain uncertainty over other agents' goals. Naively adding such uncertainty as part of the environment would add a large bias to the simulation and degrade planning performance. To overcome this problem, we propose to sample opponents' goal combinations according to the belief maintained by the opponent modeling module, and then estimate action value by MCTS based on the samples. To balance the trade-off between computational complexity and planning performance, we repeat the process multiple times and choose actions according to the average action value. In the following, we first introduce the necessary background of MCTS. We then proceed to introduce how we plan for a rewarding action under the uncertainty over opponent policies.

**MCTS.** Monte Carlo Tree Search (MCTS) is a type of tree search that plans for the best action at each time step (Silver & Veness, 2010; Liu et al., 2020). MCTS uses the environment to construct a search tree (right side of Figure 1) where nodes correspond to states and edges refer to actions. Specifically, each edge transfers the environment from its parent state to its child state. MCTS expands the search in ways (such as pUCT) that properly balance exploration and exploitation. Value and visit of every state-action (node-edge) pair are recorded during expansion (Silver et al., 2016). Finally, the action with the highest value (or highest visit) of the root state (node) is returned and executed in the environment.

**Planning under uncertain opponent policies.** Based on beliefs over opponents' goals and their goal-conditioned policies from the opponent modeling module, we run MCTS for $N_s$ rounds. In each round, other agents' goals are sampled according to the focal agent's belief over opponents' goals $b_{ij}(g_j)$. Specifically, at time $t$ in episode $K$, we sample the goal combination $\mathbf{g}_{-i} = \{g_j \sim b_{ij}^{K,t}(\cdot), j \neq i\}$. Then at every state $\tilde{s}^k$ in the MCTS tree of this round, other agents' actions $\tilde{\mathbf{a}}_{-i}$ are determined by $\tilde{\mathbf{a}}_{-i} \sim \pi_{\boldsymbol{\omega}}(\cdot|\tilde{s}^k, \mathbf{g}_{-i})$ from the goal-conditioned policy.

In each round, MCTS gives the estimated action value of the current state $Q(s^{K,t}, a, \mathbf{g}_{-i}) = V(\tilde{s}'(a))$ ($a \in A_i$), where $\tilde{s}'(a)$ is the next state after taking $\tilde{\mathbf{a}}^0_{-i} \cup a$ from $\tilde{s}^0 = s^{K,t}$.

We average the estimated action value from MCTS in all $N_s$ rounds:

$$Q_{avg}(s^{K,t}, a) = \sum_{l=1}^{N_s} Q_l(s^{K,t}, a, \mathbf{g}^l_{-i}). \tag{4}$$

Agent $i$'s policy follows Boltzmann rationality model (Baker et al., 2017):

$$\pi_{MCTS}(a|s^{K,t}) = \frac{\exp(\beta Q_{avg}(s^{K,t}, a))}{\sum_{a' \in A_i} \exp(\beta Q_{avg}(s^{K,t}, a'))}, \tag{5}$$

where $\beta \in [0, \infty)$ is rationality coefficient. As $\beta$ increases, the policy gets more rational. We choose our action at time $t$ of the episode $K$ based on $\pi_{MCTS}(a|s^{K,t})$.

Note that the effectiveness of MCTS is highly associated with the default policies and values provided to MCTS. When they are close to the optimal ones, they can offer an accurate estimate of state value, guiding MCTS search in the right direction. Therefore, following Silver et al. (2018), we train a neural network $\boldsymbol{\theta}$ to predict the policy and value functions at every state following the supervision provided by MCTS. Specifically, the policy target is the policy generated by MCTS, while the value target is the true discounted return of the state in this episode.

As for state $\tilde{s}^k$ in the MCTS, the policy function $\pi^k_{\boldsymbol{\theta}}$ guides the exploration by having an impact on the pUCT functions. The value function $v^k_{\boldsymbol{\theta}}$ estimates the return and provides the initial value of $\tilde{s}^k$ when $\tilde{s}^k$ is first reached.

The network $\boldsymbol{\theta}$ is updated based on the overall loss:
$$L(\boldsymbol{\theta}) = L_p(\pi_{MCTS}, \pi_{\boldsymbol{\theta}}) + L_v(r, v_{\boldsymbol{\theta}}), \tag{6}$$
where
$$L_p(\pi_1, \pi_2) = \mathbb{E}[-\sum_{a \in A_i} \pi_1(a|s^{K,t}) log(\pi_2(a|s^{K,t})],$$
$$L_v(r, v) = \mathbb{E}[(v(s^{K,t}) - \sum_{l=t}^{\infty} \gamma^{l-t} r_i^{K,l})^2].$$

## 5 EXPERIMENTS

### 5.1 EXPERIMENTAL SETUP

Agents are tested in three representative paradigms of SSDs: sequntial stag-hunt game (SSH), sequential snowdrift game (SS), and sequential prisoner's dilemma (SPD) (see Appendix C).

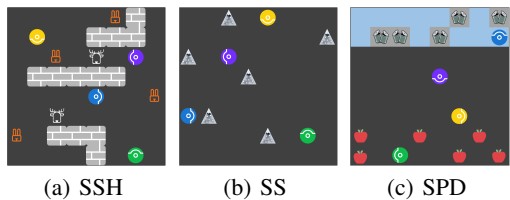

(a) SSH      (b) SS      (c) SPD

In **SSH**, four agents are rewarded for catching prey. As shown in Figure 2(a), each agent has six actions: idle, move left, move right, move up, move down, and hunt. If there are obstacles or boundaries in an agent's moving direction, its position stays unchanged. Agents can hunt prey in their current grid, and there are two types of prey: stags and hares. A stag provides a reward of 10, and requires at least two agents located at its grid to execute "hunt" together. These co-operating agents will split the reward evenly. A hare provides a reward of 1, and each agent can catch a hare alone. After a successful hunting, both the hunters and the prey disappear from the environment. The game terminates when the time $T_{max} = 30$ runs out, or terminates 5 timesteps after the first successful hunting in each episode. The dilemma in SSH is a tension between maximizing benefit (i.e., hunting stags) and minimizing risk (i.e., hunting hares). The 5-timesteps termination rule ensures that the tension between payoff-dominant cooperation and risk-dominant defection is maintained. Without this rule, agents would have enough time to hunt hares if failing to hunt a stag, and the dilemma would be diluted.

Figure 2: Overview of three representative paradigms of SSDs. There are four agents, represented by colored circles, in each paradigm. (a) Agents catch prey for reward. A stag with a reward of 10 requires at least two agents to hunt together. One agent can hunt a hare with a reward of 1. (b) Everyone gets a reward of 6 when an agent removes a snowdrift. When a snowdrift is removed, removers share the cost of 4 evenly. (c) Agents get a reward of 10 for collecting an apple, and a reward of $-1$ for cleaning a bag of waste. Apples respawn at a rate $1 - 2.5x$, where $x$ represents the percentage of waste in the river.

In **SS** (Figure 2(b)), there are six snowdrifts located randomly in an $8 \times 8$ grid. Similar to SSH, at every time step the agent can stay idle or move one step in any direction. Agents are additionally equipped with a "remove a snowdrift" action, which removes the snowdrift in the same cell as the agent. When a snowdrift is removed, removers share the cost of 4 evenly, and every agent gets a reward of 6. The game ends when all the snowdrifts are removed or the time $T_{max} = 50$ runs out. The game's essential dilemma arises from the fact that an agent can obtain a higher reward by free-riding, i.e., waiting for other agents to remove the snowdrifts, than by removing a snowdrift themselves. However, if all agents take free rides, no one will remove any snowdrifts, and the group will not receive any reward. On the other hand, if any agent is satisfied with a suboptimal strategy and chooses to remove snowdrifts, both the group benefit and individual rewards increase.

Finally, we investigate **SPD** (Figure 2(c)), inspired by the environment *Cleanup* from the Melting Pot benchmark (Leibo et al., 2021). In this $8 \times 8$ grid, there is a river in the top two rows and a forest with apples in the bottom two rows. Bags of waste are scattered throughout the river. Waste is produced in the river at a constant rate of 0.25, and the river becomes saturated with waste when it covers 40% of the river. Apples respawn at a rate of $1 - 2.5x$, where $x$ represents the percentage of waste in the river. Agents receive a reward of 10 for collecting an apple, and a reward of $-1$ for cleanup a bag of waste. The game terminates after $T_{max} = 100$ timesteps. At the beginning of each episode, the river is saturated with waste and no apple is present, so agents must consistently clean up waste to ensure the growth rate of the apple population. However, cleaning up waste hinders agents to collect apples since they are located far away in the environment. Agents receive less reward for cleaning up waste, regardless of what their opponents do, but no one receives a reward if no agents clean up waste, which is the central dilemma of SPD.

In all three environments, four agents have no access to each other's parameters, and communication between them is not allowed. Appendix D introduces the goal definition of these games.

**Baselines.** Here, some baseline algorithms are introduced to evaluate the performance of PToM. During the evaluation of few-shot adaptation, baseline algorithms serve a dual purpose. Firstly, they

act as unfamiliar opponents during the evaluation process to test the few-shot adaptation ability of PToM. Secondly, we evaluate the few-shot adaptation ability of the baseline algorithms to demonstrate PToM's superiority. *LOLA* (Foerster et al., 2018; Zhao et al., 2022) agents consider a 1-step look-ahead update of opponents, and update their own policies according to the updated policies of opponents. *SI* (Jaques et al., 2019) agents have an intrinsic reward term that incentivizes actions maximizing their influence on opponents' actions. The influence is accessed by counterfactual reasoning. *A3C* (Mnih et al., 2016) agents are trained using the Asynchronous Advantage Actor-Critic method, a well-established reinforcement learning (RL) technique. *Prosocial-A3C* (PS-A3C) (Peysakhovich & Lerer, 2018) agents are trained using A3C but share rewards between players during training, so they optimize the per-capita reward instead of the individual reward, emphasizing cooperation between players. The ablated version of PToM, *direct-OM*, retains the planning module, removes the opponent modeling module, and uses neural networks to model opponents directly (see details in Appendix F.3). In addition, we construct some rule-based strategies that are extreme strategies specific to the game. *Random* policy takes a valid action randomly at each step. An agent that consistently adopts cooperative behavior is called *cooperator*, and an agent that consistently adopts exploitative behavior is called *exploiter*. In SSH, the goals of cooperators and exploiters are hunting the nearest stag and hare, respectively. In SS, cooperators keep moving to remove the nearest snowdrift, and exploiters randomly take actions other than "remove a snowdrift". In SPD, cooperators always move to clean the nearest waste, and exploiters move to collect apples if they exist.

## 5.2 PERFORMANCE

The experiment consists of two phases. The first phase focuses on self-play training, where agents using the same algorithm are trained until convergence. Self-play ability is measured by the algorithm's average reward after convergence. The second phase evaluates the few-shot adaptation ability of PToM and learning baselines. Specifically, a focal agent interacts with three opponents using a different algorithm for 2400 steps. The focal agent's average reward during the final 600 steps is used to measure its algorithm's few-shot adaptation ability. At the start of the adaptation phase, any policy's parameters are the convergent parameters derived from the corresponding algorithms in self-play. During the phase, policies can update their parameters if possible. Implementation details are given in Appendix E. The results of self-play and that of few-shot adaptation are displayed in Table 1 and Table 2, respectively.

Table 1: Self-play performance of PToM and baseline algorithms. Shown is the normalized score after convergence in the self-play training phase.

|  | PToM | LOLA | SI | A3C | PS-A3C | direct-OM |
|---|---|---|---|---|---|---|
| SSH | **0.9767**$\pm$ 0.0117 | 0.9038$\pm$ 0.0117 | 0.9125$\pm$ 0.0233 | 0.9708$\pm$ 0.0087 | 0.7347$\pm$ 0.0029 | 0.9417$\pm$ 0.0146 |
| SS | **0.9900**$\pm$ 0.0047 | 0.6200$\pm$ 0.0070 | 0.7133$\pm$ 0.0060 | 0.6933$\pm$ 0.0113 | 0.9500$\pm$ 0.0093 | 0.7933$\pm$ 0.0080 |
| SPD | 0.0181$\pm$ 0.0012 | 0.0064$\pm$ 0.0008 | 0.0064$\pm$ 0.0005 | 0.0000$\pm$ 0.0000 | **0.4333**$\pm$ 0.0031 | 0.0163$\pm$ 0.0007 |

**SSH.** As demonstrated in Table 1, PToM and A3C perform comparably in self-play, close to the best possible reward. They both learn effective strategies that prioritize hunting stags. LOLA and SI agents have worse self-play performance than PToM and A3C. PS-A3C agents obtain the lowest reward. PS-A3C tends to delay hunting, as early hunting leads to leaving the environment and failing to obtain the group reward from subsequent hunting. Additionally, PS-A3C does not effectively learn the relationship between hunting and receiving rewards, since they can get rewards without hunting by itself. These reasons lead to PS-A3C may take suboptimal actions in the last few steps and thus fail to hunt.

PToM gains considerable returns when adapting to all other types of opponents (see Table 2(a)). Although LOLA is not as good as A3C in self-play, both have their own advantages in terms of adaptation. SI performs significantly worse than LOLA on the adaptation test, although they perform similarly in self-play. Direct-OM consistently underperforms compared with PToM across all adaptation scenarios, with some instances revealing notable disadvantages. PS-A3C, as a result of the aforementioned reasons, has fewer successful hunts, leading to inferior performance.

We would like to provide further intuition on why PToM is capable of efficiently adapting its policy to unseen agents. Take the experiment facing three exploiters (always attempting to hunt the nearest hare) as an example. There are two goals here: hunting stags or hunting hares. At the start of the

Table 2: Few-shot adaptation performance of PToM and baselines in (a) SSH, (b) SS, and (c) SPD. The interaction happens between 1 agent using the row policy and 3 other agents using the column policy. Shown are the min-max normalized scores, with normalization bounds set by the rewards of LI-Ref and the random policy. See detailed description of LI-Ref and corresponding analysis in Appendix F.1. The results are depicted for the row policy from 1800 to 2400 step.

(a) Performance in SSH

|  | learning opponents | | | | | rule-based opponents | | |
|---|---|---|---|---|---|---|---|---|
|  | PToM | LOLA | SI | A3C | PS-A3C | random | cooperator | exploiter |
| PToM | - | **0.97**± 0.02 | **0.96**± 0.03 | **0.99**± 0.02 | **0.88**± 0.02 | **0.78**± 0.07 | **1.00**± 0.01 | **0.36**± 0.03 |
| LOLA | **0.98**± 0.02 | - | 0.94± 0.02 | 0.92± 0.04 | 0.82± 0.02 | 0.75± 0.04 | **1.00**± 0.03 | 0.28± 0.03 |
| SI | 0.89± 0.02 | 0.77± 0.02 | - | 0.83± 0.01 | 0.74± 0.01 | 0.52± 0.03 | 0.87± 0.03 | 0.27± 0.02 |
| A3C | 0.96± 0.02 | 0.91± 0.02 | **0.96**± 0.03 | - | 0.87± 0.02 | 0.55± 0.05 | 0.98± 0.02 | 0.25± 0.02 |
| PS-A3C | 0.32± 0.02 | 0.24± 0.03 | 0.20± 0.03 | 0.18± 0.02 | - | 0.29± 0.02 | 0.38± 0.01 | 0.06± 0.02 |
| direct-OM | 0.86± 0.01 | 0.95± 0.01 | 0.83± 0.03 | 0.84± 0.02 | 0.74± 0.03 | 0.60± 0.04 | 0.96± 0.03 | 0.31± 0.02 |

(b) Performance in SS

|  | learning opponents | | | | | rule-based opponents | | |
|---|---|---|---|---|---|---|---|---|
|  | PToM | LOLA | SI | A3C | PS-A3C | random | cooperator | exploiter |
| PToM | - | **0.72**± 0.05 | **0.55**± 0.30 | **0.39**± 0.09 | **-0.56**± 0.39 | **0.36**± 0.03 | -1.75± 0.25 | **0.55**± 0.01 |
| LOLA | **-0.50**± 0.10 | - | -1.18± 0.31 | 0.33± 0.07 | -1.00± 0.29 | 0.14± 0.10 | -2.00± 0.20 | 0.18± 0.01 |
| SI | -0.77± 0.14 | 0.67± 0.11 | - | 0.00± 0.05 | -2.00± 0.33 | -0.14± 0.04 | -3.00± 0.35 | 0.24± 0.05 |
| A3C | -0.74± 0.15 | 0.14± 0.06 | -1.64± 0.25 | - | -1.11± 0.38 | 0.20± 0.05 | -2.50± 0.15 | 0.14± 0.01 |
| PS-A3C | -1.12± 0.11 | 0.58± 0.07 | -0.82± 0.38 | 0.35± 0.04 | - | 0.24± 0.05 | -4.25± 0.43 | 0.38± 0.02 |
| direct-OM | -0.61± 0.17 | 0.31± 0.11 | -2.46± 0.23 | 0.12± 0.05 | -0.78± 0.20 | 0.30± 0.05 | **-0.25**± 0.28 | 0.34± 0.05 |

(c) Performance in SPD

|  | learning opponents | | | | | rule-based opponents | | |
|---|---|---|---|---|---|---|---|---|
|  | PToM | LOLA | SI | A3C | PS-A3C | random | cooperator | exploiter |
| PToM | - | 1.40± 0.23 | **1.45**± 0.23 | **1.28**± 0.18 | **1.19**± 0.02 | **0.27**± 0.03 | 0.85± 0.01 | 0.84± 0.03 |
| LOLA | 0.75± 0.14 | - | 0.96± 0.09 | 1.07± 0.10 | 0.69± 0.02 | 0.05± 0.01 | **1.00**± 0.01 | 0.92± 0.06 |
| SI | 1.00± 0.11 | 0.64± 0.12 | - | 1.07± 0.06 | 0.93± 0.03 | 0.10± 0.01 | **1.00**± 0.00 | **1.00**± 0.05 |
| A3C | 0.75± 0.09 | 0.66± 0.08 | 1.00± 0.16 | - | 1.02± 0.02 | 0.17± 0.01 | **1.00**± 0.01 | **1.00**± 0.00 |
| PS-A3C | -4.32± 0.12 | -3.89± 0.12 | -4.44± 0.19 | -7.56± 0.18 | - | -0.12± 0.01 | 0.08± 0.09 | -9.25± 0.22 |
| direct-OM | **1.51**± 0.16 | **1.82**± 0.20 | 1.20± 0.15 | 1.20± 0.12 | 0.28± 0.02 | 0.20± 0.03 | 0.71± 0.03 | 0.75± 0.06 |

evaluation phase, PToM holds the belief that every opponent is more likely to hunt a stag because PToM has seen its opponents hunt stags more than hares during self-play. This false belief for exploiters degrades PToM's performance. Both intra-ToM and inter-ToM correct this false belief by updating during the interactions with exploiters (see visualization of belief update in Figure 4 in Appendix F.2). Intra-ToM provides the ability to correct the belief of hunting stags within an episode. Specifically, as an opponent keeps moving closer to a hare, intra-ToM will update the intra-episode belief for the opponent toward the goal "hare", leading to accurate opponent models. Taking these accurate opponent policies as input, the planning module can output advantageous actions. Inter-ToM further accelerates the convergence towards true belief by updating the inter-episode belief, which is used as a prior for intra-ToM at the start of every episode.

**SS.** As shown in Table 1, during self-play, PToM achieves the highest reward and it is close to the theoretically optimal average reward in this environment (i.e. when all snowdrifts are removed, resulting in a group average reward of 30.0). This outcome is a remarkable achievement in a fully decentralized learning setting and highlights the high propensity of PToM to cooperate. In contrast, LOLA, SI, and A3C prioritize maximizing their individual profits, which leads to inferior outcomes due to their failure to coordinate and cooperate effectively. PS-A3C performs exceptionally well in self-play, ranking second only to PToM. Like in SSH, it fails to achieve the maximum group average reward due to the coordination problem, which is prominent when there is only one snowdrift left. This issue highlights the instability of the strategy caused by the absence of action planning.

PToM demonstrates the most effective few-shot adaptation performance (Table 2(b)). Specifically, when adapting to three exploiters, PToM receives substantially higher rewards than other policies. This highlights the effectiveness of PToM in quickly adapting to non-cooperative behavior, which differs entirely from opponent behavior in PToM's self-play. In contrast, A3C and PS-A3C do not

explicitly consider opponents. They have learned the strategies tending to exploit and cooperate, respectively. Therefore, A3C performs effectively against agents that have a higher tendency to cooperate, such as PToM and cooperator. However, its performance is relatively poor when facing agents unlikely to cooperate. Conversely, PS-A3C exhibits the opposite behavior. Direct-OM only performs well when facing cooperators, and performs poorly when facing relatively exploitative agents such as LOLA, SI, and A3C.

**SPD.** In the scenario of decentralized training with no communication, a group of agents that optimize for their own returns can easily fall into the Nash equilibrium where individuals never clean up the waste and always try to pick apples. During self-play, PToM, along with other self-interested baselines (LOLA, SI, and A3C), converges to the equilibrium, which is attributed to the inherent characteristics of the prisoner's dilemma game (Table 1). PS-A3C agents gain high returns and escape the undesirable equilibrium to a certain extent, as they aim to maximize the collective benefit.

The adaptation results between PToM, LOLA, SI, A3C and direct-OM underscore that self-interested agents often sink into the undesirable Nash equilibrium in SPD (Table 2(c)). PToM obtains less reward than other self-interested algorithms when playing with rule-based cooperators. When faced with a new opponent, PToM tends to engage in exploratory cooperative actions to understand the opponent's characteristics. This leads to relatively lower returns for PToM. When facing an agent exhibiting dynamic behavior, such as PS-A3C, it becomes imperative for the agent to think further. In such scenarios, some apples are available, and the focal agent needs to contend with opponents for apples. It is important to choose apples to pick and plan the path for picking those apples. The planning module within PToM empowers the agent to navigate and optimize its path and thus ensures a competitive advantage. PS-A3C aims to maximize the collective average reward. Thus, it is vulnerable to exploitation by other agents, leading to low returns when playing with self-interested opponents in SPD.

Overall, this study demonstrates the remarkable adaptation ability of PToM across three distinct social dilemma paradigms. While the advantages of PToM may not be significant in specific test scenarios against particular opponents, its overall performance consistently surpasses the baselines. Meanwhile, PToM exhibits advantages during self-play.

Ablation study indicates that inter-ToM and intra-ToM play crucial roles in adapting to agents with fixed goals and agents with dynamic goals, respectively. Moreover, if opponent modeling is not conditioned on goals, the self-play and few-shot adaptation abilities are greatly weakened. Further details are provided in Appendix F.3.

We observe the emergence of social intelligence, including self-organized cooperation and an alliance of the disadvantaged, during the interaction of multiple PToM agents in SSDs. Further details can be found in Appendix G.

## 6 Conclusion and Discussion

We propose Planning with Theory of Mind (PToM), a hierarchical algorithm for few-shot adaptation to unseen opponents in SSDs. It consists of an opponent modeling module for inferring opponents' goals and behavior and a planning module guided by the inferred information to output the focal agent's best response. Empirical results in three typical SSD paradigms (SSH, SS, and SPD) show that PToM performs better than state-of-the-art MARL algorithms, in terms of dealing with complex SSDs in the self-play setting and few-shot adaptation to previously unseen opponents.

Whilst PToM exhibits superior abilities, there are several limitations illumining our future work. First, in any environment, a clear definition of goals is needed for PToM. To enhance PToM's ability to generalize to various environments, a technique that can autonomously abstract goal sets in various scenarios is needed. Second, we investigate complex SSDs with the expectation that PToM can facilitate effective decision-making and adaptation in human society. Despite selecting diverse well-established algorithms as opponents, none of them adequately model human behavior. It would be interesting to explore how PToM can perform in a few-shot adaptation scenario involving human participants. As PToM is self-interested, it may not always make decisions that are in the best interest of humans. One way to mitigate this risk is leveraging PToM's ability to infer and optimize for human values and preferences during interactions, thereby assisting humans in complex environments.

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
