# A PSEUDO CODE OF PTOM

---
**Algorithm 1** PToM
---
**Input:** Number of MCTS tree $N_s$, update interval $T_u$, capacity of the trajectory buffer $L$, goal set $G_j$ ($j \neq i$), initial belief of agents' goals $b_{ij}^{0,0}(g_j)$.

**Output:** Actions $a_i^{K,t}$, planning module network $\boldsymbol{\theta}$, goal-conditioned policy network $\boldsymbol{\omega}$.

**for** each episode $K$ **do**
    generate initial state of this episode $s^{K,0}$ randomly
    **for** $t = 0$ **to** $T_{max} - 1$ **do**
        **repeat**
            sample $\mathbf{g}_{-i}^l$ from $b_{ij}^{K,t}(g_j)(j \neq i)$
            get $Q_l(s^{K,t}, a, \mathbf{g}_{-i}^l)$ ($\forall a$) via MCTS
        **until** $N_s$ times
        calculate $Q_{avg}(s^{K,t}, a)$ ($\forall a$) [Eq. 4]
        choose action $a_i^{K,t}$ from $\pi_{MCTS}(a|s^{K,t})$ [Eq. 5]
        intra-ToM update $b_{ij}^{K,t+1}$ [Eq. 1]
        collect data of this step to the trajectory buffer
    **end for**
    **if** the trajectory buffer is full **then**
        update $\boldsymbol{\omega}$ [Eq. 3]
    **end if**
    **if** $K \times T_{max} \equiv 0 \pmod{T_u}$ **then**
        update $\boldsymbol{\theta}$ [Eq. 6]
    **end if**
    inter-ToM update $b_{ij}^{K+1,0}$ [Eq. 2]
**end for**

---

# B THEORETICAL ANALYSIS

We aim to offer a concise theoretical analysis. Due to the complexity of environments characterized by both temporal and spatial structures, attaining theoretical guarantees in such environments can be inherently challenging. To strike a balance, we have undertaken a verification of the theoretical guarantee associated with PToM in the matrix games. These games encapsulate the same dilemma of sequential games. For clarity, our analysis will be conducted in the context of a two-player game, and the analysis can be extended to games involving a greater number of agents. Consider a two-player game where both players have two goals: "Cooperate" and "Defect," resulting in a utility matrix shown in Table 3.

Table 3: Utility matrix for a two-player game. Each element in the table represents the utility of the row player (first value) and the utility of the column player (second value). The utility values $R$, $S$, $T$, and $P$ determine different game paradigms.

|           | Cooperate | Defect |
|-----------|-----------|--------|
| Cooperate | $R, R$    | $S, T$ |
| Defect    | $T, S$    | $P, P$ |

Suppose PToM is the row player. At a certain timestep, the column player selects its goal $g_{column}$ to be "Cooperate" with a probability of $p$ and to be "efect" with a probability of $1 - p$. We sample the opponent's goal to simulate using Monte Carlo Tree Search (MCTS), with a frequency of $p + \epsilon$ to "Cooperate" and a frequency of $1 - p - \epsilon$ to "Defect."

In the current state $s$, we have two possible actions: $a_1$ for cooperation and $a_2$ for defection. During the MCTS planning process, when the opponent aims to "Cooperate," we have:

$$Q(s, a_1 | g_{\text{column}} = \text{"Cooperate"}) = R(1 + \epsilon_R)$$
$$Q(s, a_2 | g_{\text{column}} = \text{"Cooperate"}) = T(1 + \epsilon_T)$$

When the opponent aims to "Defect," we have:

$$Q(s, a_1 | g_{\text{column}} = \text{"Defect"}) = S(1 + \epsilon_S)$$
$$Q(s, a_2 | g_{\text{column}} = \text{"Defect"}) = P(1 + \epsilon_P)$$

Thus, we can calculate the overall Q-values as follows:

$$Q(s, a_1) = (p + \epsilon)R(1 + \epsilon_R) + (1 - p - \epsilon)S(1 + \epsilon_S)$$
$$Q(s, a_2) = (p + \epsilon)T(1 + \epsilon_T) + (1 - p - \epsilon)P(1 + \epsilon_P)$$

In the learning process, the goal-conditioned policy network is trained using supervised learning, and its accuracy significantly improves with sufficient rounds of observation. Consequently, the accuracy of the environment simulation within the Monte Carlo Tree Search (MCTS) algorithm becomes exceedingly high. In such a scenario, the convergence guarantee of MCTS remains intact, resulting in a final precision of MCTS that is remarkably high. Specifically, we have $|\epsilon_R|, |\epsilon_S|, |\epsilon_T|, |\epsilon_P| \ll |\epsilon|$, and these small error terms can be safely ignored.

Then, when

$$\frac{T + S - R - P}{p(R - T) + (1 - p)(S - P)}\epsilon < 1,$$

the optimal strategy that PToM obtains is consistent with the true optimal strategy. Two factors affect the size of $|\epsilon|$: the accuracy of ToM in inferring the opponent's goals and the deviation between frequency and probability when sampling the goal. To address the accuracy issue, we employ two layers of modules, intra-ToM and inter-ToM, to make accurate predictions as early as possible in each episode. For the deviation between frequency and probability, we increase the value of $N_s$ to reduce this deviation. In practical applications, the choice of an appropriate $N_s$ depends on the trade-off between computational speed and sampling accuracy.

## C    SCHELLING DIAGRAMS OF SSDS

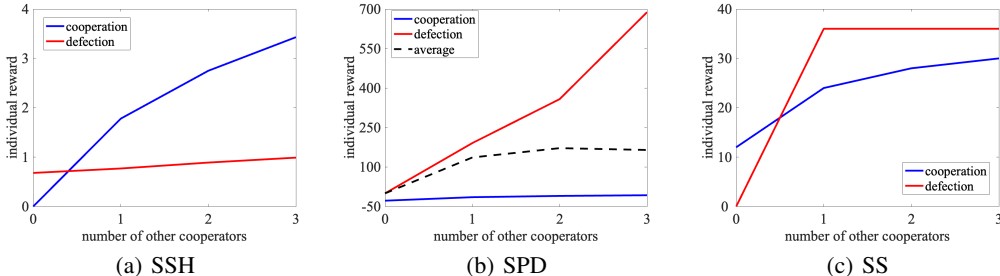

(a) SSH            (b) SPD            (c) SS

Figure 3: Schelling diagrams for (a) SSH, (b) SPD, and (c) SS. The black dashed line in (b) shows the overall average return were the individual to choose defection.

Game types are determined by the relative values of elements in the payoff matrix. The Schelling diagram compares the rewards of different potential strategies (i.e., cooperation and defection here) given a fixed number of other cooperators. It is a natural generalization of the payoff matrix for two-player games to multi-player settings. Here, we use Schelling diagrams to validate our temporal and spatial extension of the matrix-form games.

Figure 3(a) shows the Schelling diagram of SSH. Defection (i.e., hunting hare) is a safe strategy as a reasonable reward is guaranteed independent of the other agents' strategies. Cooperation (i.e., hunting stag) poses the risk of being left with nothing (when there are no others hunting stag), but is more rewarding if at least one other agent hunts stag. That is to say, hunting hare is risk dominant, and hunting stag is reward dominant. This is consistent with the dilemma described by the matrix-form stag-hunt game (Bloembergen et al., 2011).

Prisoner's dilemma describes the tension between self-interest and group benefit. For a player, the optimal strategy is defection independent of others' strategies, which brings about the only

Nash equilibrium where everyone defects. However, from the perspective of the whole group, total defection leads to minimal group benefit. Instead, group benefit is maximized by mutual cooperation. Figure 3(b) shows that defection (i.e. collecting apples) is always a better choice than cooperation (i.e., picking up waste) and that the average reward increases with the number of cooperators. This demonstrates that SPD keeps the nature of the prisoner's dilemma.

Different from the prisoner's dilemma where cooperation incurs cost to the acting players and benefits only others, in the snowdrift game, the costly cooperation can accrue benefit not only to others but also to the acting player (Souza et al., 2009). For the snowdrift game, there are two pure-strategy Nash equilibria: player 1 cooperates and player 2 defects; player 1 defects and player 2 cooperates. That is, the optimal strategy is playing the strategy different from the coplayer. As shown in Figure 3(c), in SS, one player's optimal strategy is cooperation (i.e. removing snowdrifts) when no other players cooperate, but when there are other cooperators, the optimal strategy is defection (i.e., free-riding). Our SS is an appropriate extension of the matrix-form snowdrift game.

# D  GOAL DEFINITION IN SSDS

## D.1  SEQUENTIAL STAG-HUNT GAME (SSH)

In SSH, we define two goals: $g^C$ as hunting stags and $g^D$ as hunting hares.

## D.2  SEQUENTIAL SNOWDRIFT GAME (SS)

In SS, we define two goals: $g^C$ as removing the drifts, and $g^D$ as staying lazy (i.e. not attempting to remove any snowdrifts). For inter-ToM, the goal $g^C$ is decomposed into 6 parts: $g^{Ck}$ ($1 \leq k \leq 6$), where $g^{Ck}$ represents removing $k$ snowdrift(s) in one episode. $b_{ij}^{K,0}(g^{Ck})$ and $b_{ij}^{K,0}(g^D)$ will be updated according to Eq. 2. During an episode, if the opponent $j$ has removed $m$ snowdrift(s) at time $t$ of the episode $K$, our belief $b_{ij}^{K,t}(g_j^C) = \sum_{k=m+1}^{6} b_{ij}^{K,0}(g_j^{Ck})$.

For intra-ToM, each snowdrift $s$ is defined as a subgoal $g^{C[s]}$. We use Eq. 1 conditioned on $g^C$ to update our belief:

$$b_{ij}^{K,t+1}(g_j^{C[s]}|g_j^C) = \frac{1}{Z_1} b_{ij}^{K,t}(g_j^{C[s]}|g_j^C) Pr_i(a_j^{K,t}|s^{K,0:t}, g_j^{C[s]}),$$

where $Z_1$ is the normalization factor. We can update our belief of an agent removing a snowdrift $s$:

$$b_{ij}^{K,t}(g^{C[s]}) = b_{ij}^{K,t}(g_j^{C[s]}|g_j^C) b_{ij}^{K,t}(g_j^C).$$

At the start of an episode, $b_{ij}^{K,0}(g_j^{C[s]}|g_j^C)$ is set to be uniform, which means $b_{ij}^{K,0}(g_j^{C[s]}|g_j^C) = \frac{1}{6}$. We train the goal-conditioned policy network $\boldsymbol{\omega}$ conditioned on $g^{C[s]}$.

## D.3  SEQUENTIAL PRISONER'S DILEMMA (SPD)

In SPD, we define two goals: $g^C$ as cleaning up waste and $g^D$ as collecting apples.

Since each goal can be achieved multiple times in a single episode, PToM updates its prior belief $b_{ij}^{K,0}$ at every time when agent $j$ achieves a goal. At the end of the current episode, this value will be assigned to the prior of the next episode, i.e., $b_{ij}^{K+1,0}$.

# E  IMPLEMENTATION DETAILS

## E.1  MCTS SIMULATION DETAILS

As introduced in Sec. 4.2, we run MCTS for $N_s$ rounds. In each round, we run $N_i$ search iterations (see Browne et al. (2012) for details of each iteration). The score of an action $a$ at state $\tilde{s}^k$ is:

$$Score(\tilde{s}^k, a) = \text{sign}(Q(\tilde{s}^k, a)) \log(1 + |Q(\tilde{s}^k, a)|) + c\pi_{\boldsymbol{\theta}}(a|\tilde{s}^k) \frac{\sqrt{\sum_{a'} N(\tilde{s}^k, a')}}{N(\tilde{s}^k, a)}$$

where $Q(\tilde{s}^k, a)$ denotes the average return obtained by selecting action $a$ at state $\tilde{s}^k$ in the previous search iterations. $N(\tilde{s}^k, a)$ represents the number of times action $a$ has been selected at state $\tilde{s}^k$ in the previous search iterations. $\pi_{\boldsymbol{\theta}}(a|\tilde{s}^k)$ refers to the policy provided by the network $\boldsymbol{\theta}$. $c$ is the exploration coefficient. The sign function $\text{sign}(\cdot)$ returns a value of 1 if the input is non-negative, and -1 if the input is negative. We select the action which has the highest score when reaching $\tilde{s}^k$ at the selection phase of one search iteration.

### E.2 NETWORK ARCHITECTURE

The goal-conditioned policy network $\boldsymbol{\omega}$ and the policy-value network for MCTS $\boldsymbol{\theta}$ both start with three convolutional layers with the kernel size 3 and the stride size 1. Three layers have 16, 32, and 32 output channels, respectively. They are connected to two fully connected layers. The first layer has an output of size 512, and the second layer gives the final output.

### E.3 HYPERPARAMETERS

For each result in Table 1, Table 2, and Table 6, we performed 10 independent experiments using different random seeds. The left-hand side of $\pm$ represents the average reward of the 10 trials, and the right-hand side represents the standard error.

Hyperparameters for PToM are listed in Table 4.

Table 4: Hyperparameters in PToM

|  | SSH | SS | SPD |
| --- | --- | --- | --- |
| horizon weight $\alpha$ | 0.99 | 0.99 | 0.99 |
| rationality coefficient $\beta$ | 2 | 2 | 2 |
| discount factor $\gamma$ | 0.95 | 0.95 | 0.95 |
| update interval $T_u$ | 2000 | 2000 | 2000 |
| capacity of the trajectory buffer $L$ | 5000 | 5000 | 5000 |
| number of MCTS rounds $N_s$ | 8 | 5 | 4 |
| number of search iterations for each MCTS $N_i$ | 200 | 200 | 200 |
| exploration coefficient $c$ | 3 | 6 | 4 |

## F SUPPLEMENTARY RESULTS

### F.1 LONG INTERACTION REFERENCE

To compare and evaluate the performance of few-shot adaptation between PToM and learning baselines, we train a Long Interaction Reference (LI-Ref) agent to see how well a well-established RL agent can perform in adaptation to opponents through extensive interactions

Specifically, for every type of opponents, one LI-Ref interacts with them and is trained via A3C to converge from scratch. During the training phase, opponents' parameters are fixed, which are the convergent parameters in their self-play. In the subsequent adaptation phase, the trained LI-Ref agent is tested in the same way as PToM and baseline algorithms. This process ensures that the LI-Ref agent engages in extensive interactions with the agents it would encounter during the adaptation phase. We use the LI-Ref agent's performance in the adaptation phase as a reference point to explain PToM's performance.

PToM performs close to LI-Ref across the majority of scenarios, except when random agents or exploiters serve as the opponent. This divergence primarily arises from the stability of the opponent's behavior in such scenarios, resulting in a relatively consistent optimal policy throughout each episode. Over an extended duration of interaction, LI-Ref effectively acquires a robust and high-quality policy. Surprisingly, PToM can even surpass LI-Ref in SPD especially when facing dynamic opponents like PS-A3C. In such scenarios, the optimal response behavior may change dynamically along with the opponent's behavior in an episode. The ability to dynamically model opponents and plan accordingly within the episode becomes imperative. Such ability may not be entirely compensated by prolonged interaction experience.

Table 5: Few-shot adaptation performance of LI-Ref in all three sequential social dilemma paradigms. The interaction happens between 1 LI-Ref agent and 3 other agents using the column policy. Shown is the average reward for LI-Ref from $1800$ to $2400$ step.

| | learning opponents | | | | | rule-based opponents | | |
|---|---|---|---|---|---|---|---|---|
| | PToM | LOLA | SI | A3C | PS-A3C | random | cooperator | exploiter |
| SSH | 3.47± 0.02 | 3.23± 0.02 | 3.40± 0.02 | 3.46± 0.01 | 3.97± 0.02 | 1.22± 0.01 | 3.42± 0.02 | 0.70± 0.01 |
| SS | 31.0± 0.14 | 20.9± 0.12 | 23.3± 0.11 | 22.7± 0.17 | 32.5± 0.12 | 16.0± 0.08 | 36.0± 0.00 | 12.0± 0.00 |
| SPD | 1.6± 0.28 | 2.0± 0.46 | 1.3± 0.29 | 0.0± 0.02 | 170.4± 5.56 | 94.2± 1.76 | 691.0± 0.88 | 0.0± 0.00 |

## F.2 VISUALIZATION OF BELIEF UPDATE

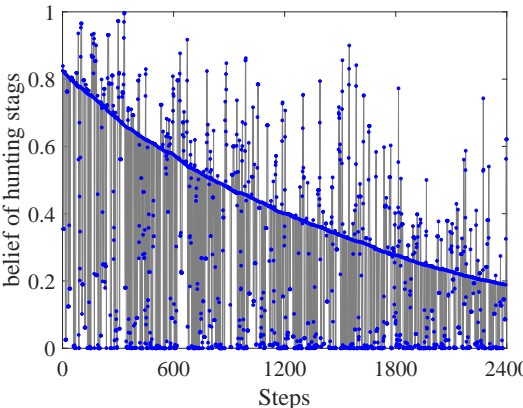

Figure 4: Visualization of PToM's belief in adaptation to one exploiter in SSH. Every blue-filled circle represents PToM's inferred probability (i.e., belief) that an opponent hunts stags. The grey line is a visual guide.

Figure 4 shows the update of a PToM agent's belief over its opponent's goal when it faces three previously unseen exploiters in SSH. In the training phase, four PToM agents interact with each other. Their beliefs over opponents' goals converge at hunting stags with a high probability. Thus, at the start of the adaptation phase shown in Figure 4, PToM's belief of hunting stags is high. During the interactions with these previously unseen opponents, PToM keeps updating its belief. PToM's belief constantly approaches the true goal of opponents, which is hunting hares.

## F.3 ABLATION STUDY

To test the importance and necessity of each component in PToM, we construct three partially ablated versions of PToM. The agent without inter-ToM (w/o inter-ToM) does not execute the inter-episode update expressed as Eq. 2. W/o inter-ToM begins each episode with a uniform belief prior. The agent without intra-ToM (w/o intra-ToM) does not execute the intra-episode update expressed as Eq. 1. That is, for w/o intra-ToM, $b_{ij}^{K,t}(g_j) = b_{ij}^{K,0}(g_j), \forall t$. The direct-OM agent removes the whole opponent modeling module of PToM, and utilizes neural networks to model opponents directly. The opponent policies are mappings from states to actions, and not conditioned on goals. Experimental results for PToM and its three ablation versions in SSH are shown in Table 6.

In self-play, PToM have an advantage over direct-OM agents. It suggests that utilizing a goal as a high-level representation of agents' behavior is beneficial to opponent modeling in complex environments. On the other hand, compared with w/o inter-ToM and w/o intra-ToM, PToM does not exhibit a significant advantage in self-play. The inter-ToM and intra-ToM modules may not be effective in the self-play setting, where a large number of interactions happen.

In the experiments testing few-shot adaptation, PToM outperforms its ablation versions. W/o inter-ToM agents struggle when facing agents with fixed goals, such as cooperators and exploiters. As the goals of cooperators and exploiters are fixed, correct actions can be taken immediately if the focal

Table 6: Performance of PToM and its ablation versions in SSH. In (a) self-play, 4 agents of the same kind are trained to converge. Shown is the normalized score after convergence. In (b) few-shot adaptation, the interaction happens between 1 agent using the row policy and 3 other agents using the column policy. Shown are the min-max normalized scores, with normalization bounds set by the rewards of LI-Ref and the random policy. The results are depicted for the row policy from 1800 to 2400 step.

(a) Self-play performance

| PToM | w/o inter-ToM | w/o intra-ToM | direct-OM |
|---|---|---|---|
| $\mathbf{0.9767}_{\pm 0.0117}$ | $0.9708_{\pm 0.0146}$ | $0.9738_{\pm 0.0117}$ | $0.9417_{\pm 0.0146}$ |

(b) Few-shot adaptation performance

| | learning opponents | | | | rule-based opponents | | |
|---|---|---|---|---|---|---|---|
| | LOLA | SI | A3C | PS-A3C | random | cooperator | exploiter |
| PToM | $\mathbf{0.97}_{\pm 0.02}$ | $\mathbf{0.96}_{\pm 0.03}$ | $\mathbf{0.99}_{\pm 0.02}$ | $\mathbf{0.88}_{\pm 0.02}$ | $\mathbf{0.78}_{\pm 0.07}$ | $\mathbf{1.00}_{\pm 0.01}$ | $\mathbf{0.36}_{\pm 0.03}$ |
| w/o inter-ToM | $\mathbf{0.97}_{\pm 0.02}$ | $0.95_{\pm 0.02}$ | $0.92_{\pm 0.03}$ | $0.87_{\pm 0.02}$ | $\mathbf{0.78}_{\pm 0.03}$ | $0.96_{\pm 0.02}$ | $0.31_{\pm 0.02}$ |
| w/o intra-ToM | $0.95_{\pm 0.02}$ | $0.94_{\pm 0.03}$ | $0.98_{\pm 0.02}$ | $0.84_{\pm 0.01}$ | $0.65_{\pm 0.04}$ | $0.99_{\pm 0.02}$ | $0.34_{\pm 0.03}$ |
| direct-OM | $0.95_{\pm 0.01}$ | $0.83_{\pm 0.03}$ | $0.84_{\pm 0.02}$ | $0.74_{\pm 0.03}$ | $0.60_{\pm 0.04}$ | $0.96_{\pm 0.03}$ | $0.31_{\pm 0.02}$ |

agent has accurate goal priors. W/o inter-ToM agents lack accurate goal priors at the beginning of an episode. In every episode, they have to use multiple interactions to infer opponents' goals and thus miss out on early opportunities to maximize their interests.

W/o intra-ToM agents exhibit poor performance when facing agents with dynamic behavior such as LOLA, SI, PS-A3C, and random. These opponents have multiple goals, which all have a non-zero probability of being chosen. But in a given episode, the specific goals of an opponent can be gradually determined by analyzing its trajectory in this episode. However, w/o intra-ToM agents can only count on inter-ToM, which only takes the past episodes into account, but does not consider the information from the current episode. It results in inaccurate goal estimates in a given episode, which hurts the performance in few-shot adaptation.

Direct-OM agents are at an overall disadvantage. Their opponent modeling solely relies on the neural network, which makes it challenging to obtain significant updates during a short interaction. This leads to inaccurate opponent modeling during the adaptation phase. Furthermore, direct-OM agents utilize end-to-end opponent modeling, which introduces a higher degree of uncertainty compared to the goal-conditioned policy. This uncertainty can reduce the precision of the simulated opponent behavior during planning.

## G    EMERGENCE OF SOCIAL INTELLIGENCES

There are two kinds of social intelligence, self-organized cooperation and the alliance of the disadvantaged, emerging from the interaction between multiple PToM agents in SSH. We make a minor modification to the game: the game terminates only when the time $T_{max} = 30$ runs out.

**Self-organized cooperation.**    As shown in Figure 5(a), at the start of the game, three agents (blue, yellow, and purple) are two steps away from the stag at the bottom-right side, and the last agent (green) is spawned alone in the upper left corner. One simple strategy for the three agents located at the bottom-right corner is to hunt the nearby stag together. Although this is a riskless strategy, the three agents each only obtain a reward of $10/3$. Instead, if one agent chooses to collaborate with the green agent at the top-left corner, all four agents each get a reward of $5$. This strategy is riskier since if the green agent chooses to hunt a nearby hare, the collaborative agent will not be able to catch any stag. We show that PToM is able to achieve the aforementioned risky but rewarding collective strategy. Specifically, the green agent refuses to catch the hare at his feet and shows the intention of cooperating with others (see screenshots at step 3 and step 8 in Figure 5(a)). The yellow agent refuses to catch the stag at the bottom-right corner and chooses to collaborate with the green agent to hunt the stag in the top-left corner. In this process, all four agents receive the maximum profit.

Here, agents achieve pairwise cooperation through independent decision-making, without centralized assignment of goals. Thus, we call this phenomenon self-organized cooperation.

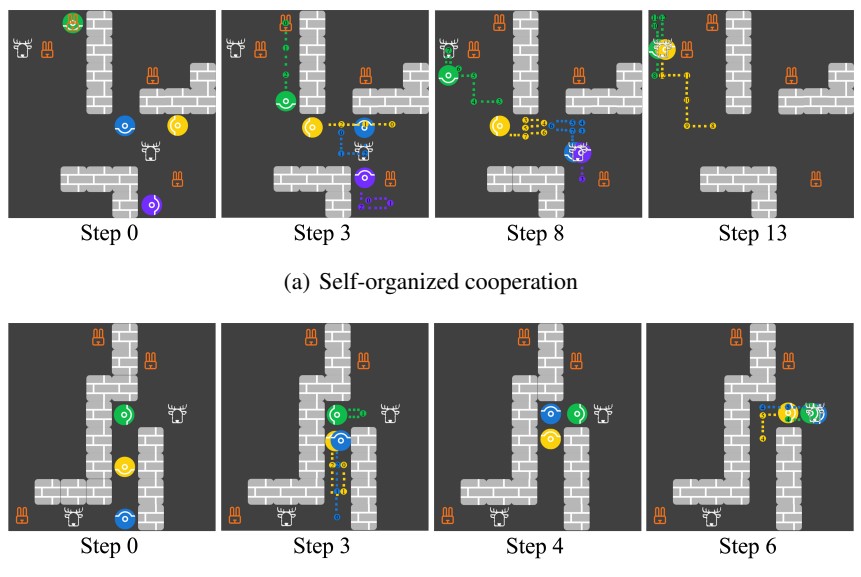

(a) Self-organized cooperation

(b) Alliance of the disadvantaged

Figure 5: Screenshots for the emergence of (a) self-organized cooperation and (b) alliance of the disadvantaged. Each panel shows agents' locations at the current step and the trajectories between the current step and the previously stated step.

**Alliance of the disadvantaged.** In addition to the aforementioned game rules, we assume agents are heterogeneous. Specifically, the yellow agent (Y) is three times greedier than the blue agent (B) and the green agent (G). That is, when the three agents cooperate to hunt a stag successfully, Y will get a reward of 6, and the others get 2 each. When Y cooperates with one of B and G, Y will obtain 7.5, the other one gets 2.5. As shown in Figure 5(b), at the start of the game, Y locates between B and G. Neither B nor G would like to cooperate with Y. Hence they need to move past Y to cooperate with each other. To achieve this, agents B and G first move closer to each other in the first few steps. However, to maximize its own profit, agent Y also moves toward B and G and hopes to hunt a stag with them. To avoid collaboration with agent Y, after agents B and G are close enough to each other, they move back and forth to mislead Y (see step 3 of Figure 5(b)). Once agent Y makes a wrong guess of the directions agents B and G move, B and G will get rid of Y, and move to the nearest stag to achieve cooperation (see Step 4 and 6 of Figure 5(b)), which maximizes the profit of agents B and G.

From the above two cases, we find that although PToM aims to maximize self-interest, cooperation emerges from the interaction between multiple PToM agents in SSDs. This shows that it may be helpful in solving SSDs by equipping agents with the ability to infer others' goals and behavior and the ability to fast adjust their own responses.