# OpenReview forum: "Planning with Theory of Mind for Few-Shot Adaptation in Sequential Social Dilemmas"
_ICLR.cc/2024/Conference — Submitted to ICLR 2024_

### Official Review · Reviewer_v2eY · 2023-10-31

**Soundness:** 3 good
**Presentation:** 3 good
**Contribution:** 3 good
**Rating:** 6
**Confidence:** 4

**Summary:**

This paper introduces a model-based reinforcement learning algorithm for strategic decision making in sequential social dilemmas (i.e. Markov games) called Planning with Theory-of-Mind (PToM). PToM agents learn to flexibly respond to other agents by inferring their goals from their actions (an aspect of theory-of-mind), then approximating a best response to those agents via Monte Carlo Tree Search (MCTS). Within each episode, goals beliefs are updated via Bayesian inference using a learned opponent model that assigns a likelihood to each action an opponent might taken given their current goal and state (intra-ToM). These opponent models are also used to simulate how the other agents will act during MCTS. Goal beliefs are also updated across episodes by taking a weighted average of the most recent goal belief and the goal belief for earlier episodes (inter-ToM). The authors show that by explicitly performing inference over opponent goals and responding accordingly via planning, PToM outperforms baseline (model-free) multi-agent RL algorithms in both self play (as measured by convergence rate and the reward achieved after convergence) and few-shot adaptation to other algorithms (as measured by performance after 2400 episode steps). They also show that PToM outperforms ablated algorithms without goal updating or opponent modeling, and demonstrates emergent cooperative behavior.

**Strengths:**

This paper takes what I think of as the "right approach" to building agents that can rapidly adapt to the policies of other agents in the few-shot and online settings: Explicitly model the goals (or to use game-theoretic language, "types") of other agents, as well as their (approximately rational) goal-directed policies, then use Bayesian inference to rapidly infer the goals/types of those agents from their actions. While this approach comes with a number of representational commitments -- e.g. the need to specify what goals other agents might have -- it also defines a normative standard (or something close to it) that basically any intelligent agent would have to approximate in order to rapidly and flexibly coordinate with other agents in the environment: Such an agent would inevitably have to learn the types of the other agents it is playing against in order to best respond to them, and the right way to learn those types from just a few actions or episodes is Bayesian inference.

PToM appears to successfully approximate that normative standard in the multi-agent RL environments studied by the authors, integrating Bayesian goal inference (as an aspect of theory-of-mind), neural opponent modeling, and model-based planning with neurally-guided MCTS in order to achieve high performance in both self-play and few-shot adaptation to unseen agents, while also showing intuitively cooperative behavior within single episodes. While I have several questions about how exactly the goal-conditioned policies for the opponents are trained, overall PToM appears to be a coherent and well-motivated approach to integrating learned neural components with explicit model-based reasoning about the goals of other agents. In particular, it seems to me that the various components should bootstrap each other in a virtuous manner during the course of self-play: Since the learned goal-directed policies try to imitate the actions produced by MCTS, they should converge to reasonably policies quite quickly since the actions produced by model-based search are going to be well-informed. In turn, improvements in the opponent model lead to more informative MCTS rollouts, which in turn improve the value and policy networks used to guide MCTS.

Empirically, PToM largely outperforms the MARL baselines tested by the authors. In a way, this is unsurprising, since there's no reason to suspect that these offline trained RL baselines would be able to succeed in few-shot adaptation and within-episode coordination with novel agents without an online learning mechanism (e.g. explicit Bayesian goal inference, or a meta-learning approximation of such inference). Nonetheless, the results clearly demonstrate the value of imbuing agents with the ability to perform online inference of the types of their opponents, in contrast to the offline focused paradigm that is common in MARL. The ablation results also validate the importance of individual PToM components, showing in particular that vanilla opponent modeling is not enough to achieve the same level of performance compared to opponent modeling combined with online goal inference.

**Weaknesses:**

Like many other model-based RL algorithms (e.g. AlphaZero), a limitation faced by PToM is that it requires additional representational commitments compared to purely model-free methods (e.g. A3C). In particular, PToM requires that agents have a veridical model of the environment to plan over via MCTS, and, as noted by authors, also requires some specification of the space of possible goals that other agents might have.

That said, this is not a huge limitation to my mind --- specifying the space of possible goals is not much of an additional assumption, once one is willing to assume that PToM agents also have accurate models of their environment. Furthermore, these representational assumptions seem like they can be addressed by future research, similar to how MuZero lifted the need for agents to have accurate simulators for MCTS. (One could also argue that PToM's model-based approach is preferable for interpretability and safety reasons, compared to a completely learned but uninterpretable world model and goal space.)

To me, the primary limitation of this paper is more methodological --- while Figure 3 shows how PToM converges much more rapidly during self-play as compared to other methods, and Table 1 shows how PToM performs better in terms few-shot adaption, it is not clear to me how much of this improvement is due to the fact that PToM uses model-based planning via MCTS, versus the fact that PToM leverages goal inference over other agents. Intuitively, it seems to me that any model-based RL algorithm would converge a lot faster during self-play (and also adapt faster), as compared to the model-free MARL baselines that PToM is compared against. Thus, I think it's important to add a model-based RL baseline to both Figure 3 and Table 1, in order to better evaluate how much the ToM component is improving performance, vs. model-based planning alone. Perhaps the direct-OM ablation in the Supplementary Results could play this role -- or as an alternative, it'd be interesting to see the performance of neurally guided MCTS when the other opponents are assumed to act in a complete random manner.

Speaking of planning, one issue that came to mind is that in general, it seems like agents should be performing *belief-space planning*, given that they have uncertainty over their opponents' goals, not just planning in the original state space of the MDP. While sampling possible goal combinations from the goal belief provides for some amount of uncertainty-awareness, it does not account for the possibility of information gathering actions that reduce uncertainty about the opponents' goals. Given the difficulty of belief-space planning, I don't expect the authors to implement that as an alternative, but I do think belief-space planning should be mentioned as a potential avenue of future work.

Regarding the clarity of the paper, I found the description and exposition mostly clear, apart from what seem to be important missing details about how the opponent module is trained, and how the replay buffer for them is collected. In particular, Eq. (3) suggests that in order to train a goal-directed policy for each goal $g$ that an agent might have, a replay buffer containing the state, action, and goal history is needed. But how is the goal history collected, given that there are no "ground truth" goals in the environment simulator? Is this goal history imputed somehow? Or are the goals generated via sampling at the start of each MCTS round? It wasn't clear to me what the answer was, even after looking at the PToM pseucode.

As a final, less important, weakness, the inter-ToM belief update shown in Eq. 2 seems rather ad hoc right now. While taking a weighted average of current and past goal beliefs seems like it makes sense intuitively, I believe this can be better justified as a Bayesian update of the goal belief across episodes, under a model where there's some probability that the opponent will resample its goal from the uniform prior at each timestep. I would encourage the authors to motivate or justify Eq. 2 in those terms. I believe this will also alleviate the requirement of "knowing" what the true goal of each agent is at the end of the episode (which Eq. 2 seems to assume), since it is possible to perform the inter-episode Bayesian update without knowing the true goal.

**Questions:**

1. How does PToM perform compared with an agent that performs neurally-guided MCTS trained via self-play like AlphaZero, and how much of the fast convergence benefit comes from model-based planning vs. ToM?

2. How is the replay buffer for opponent goals collected, given that there are no ground-truth goals?

3. Can the inter-ToM update in Eq. (2) be justified as a Bayesian update with respect to some model of how goals change over time?

4. Have the authors considered using belief-space planning (e.g. POMCP) instead of MCTS for planning under uncertainty?

5. In the Problem Formulation section, I believe the appropriate formalism here is not MDPs, but Markov Games [1]

6. In Eq. 5, is there a reason to use a Boltzmann model here, instead of just always picking the best action estimated by MCTS? Is this meant to encourage exploration?

7. In Appendix E.3, I would avoid using the term "partially crippled", since "cripple" is often considered pejorative. "Ablated" would be a good replacement.

8. Why does LOLA do better at converging than PToM in SPD, as seems to be shown by Figure 4(b)?

[1] Liu, Q., Szepesvári, C., & Jin, C. (2022). Sample-efficient reinforcement learning of partially observable Markov games. Advances in Neural Information Processing Systems, 35, 18296-18308.

---

> ### Author Response · Authors · 2023-11-18
> **Response to Reviewer v2eY (Part 1)**
>
> > Baselines
>
> We have tried a method of planning through self-play like AlphaZero. However, its computational complexity proved prohibitively high. The number of nodes required for expansion in MCTS increases exponentially with the increasing number of agents. It poses a significant chanllenge. Given our resource constraints, ensuring efficient training speed became impractical. Thus, we forwent AlphaZero as a direct baseline. A similar rationale precluded the adoption of MuZero.
>
> An alternative approach to implement self-play involves substituting one's own policy network for the opponent modeling network in direct-OM. In the symmetric games, which are studied by this paper, this approach is similar to direct-OM. However, it may not be feasible to apply this approach in the environments characterized by asymmetric player roles and rewards, Like battle of the sexes, another classic paradigm of social dilemmas.
>
> We use the direct-OM ablation as a model-based baseline. Direct-OM performs MCTS based on opponent modeling instead of self-play.From the results of direct-OM, we believe fast convergence mainly comes from model-based planning. As the baselines in our comparison are predominantly model-free methods, we acknowledge the unfairness of directly comparing the number of convergence samples. Consequently, we have opted to remove the corresponding statements and figures from the paper. Nevertheless, we believe it is crucial to emphasize the merit of applying model-based methods to improve sample efficiency in the multi-agent context.
>
>
>
> > Inter-ToM update
>
> We are very grateful for this valuable suggestion. Based on Bayesian theorem, Eq. (2) can be modified as
>
> $$ b_{ij}^{K,0}(g_j)  = \frac{1}{Z_2}[\alpha b_{ij}^{K-1,0}(g_j) + (1-\alpha) p(g_j| \tau)]  = \frac{1}{Z_2}[\alpha b_{ij}^{K-1,0}(g_j) + (1-\alpha) \frac{p(\tau|g_j)p(g_j)}{p(\tau)}] \\\\
> = \frac{1}{Z_2}[\alpha b_{ij}^{K-1,0}(g_j) + (1-\alpha) \frac{p(s_0)p(g_j)\prod_{t=0}^{T_{max} - 1} \pi_{\theta_j}(a_t|s_t, g_j)p(s_{t+1}|s_t, a_t) }{p(s_0)\sum_k p(g_k)\prod_{t=0}^{T_{max} - 1} \pi_{\theta_k}(a_t|s_t, g_k)p(s_{t+1}|s_t, a_t)}] \\\\
> = \frac{1}{Z_2}[\alpha b_{ij}^{K-1,0}(g_j) + (1-\alpha)b_{ij}^{K-1,T_{max}}(g_j)], $$
>
> where $\tau = (s^{K-1, 0}, a^{K-1, 0}, s^{K-1, 1}, a^{K-1, 1}, ..., s^{K-1, T_{max} - 1}, a^{K-1, T_{max} - 1}, s^{K-1, T_{max}})$ is a trajectory in episode $K-1$, $\pi_{\theta_j}$ is the goal-conditioned policy of the goal $g_j$ parameterized by $\theta_j$. Base on the modified equation, the update of inter-ToM does not require knowing other agents' goals at the end of episodes. Instead, the goal inferred by intra-ToM at the last timestep in episode $k-1$ is used to update inter-ToM.
>
>  During the few-shot adaptation phase, where $b_{ij}^{K-1,T_{max}}(g_j)$ is accurate, above modification does not compromise performance. (We have made a visualization for belief update in SSH in our revised manuscript, which can support this statement.) However, considering that goals are also used in goal-conditioned policy training, the precise impact of these changes in the training stage remains uncertain.
>
> We think that the weighted average of current and past goal beliefs, shown in Eq.(2), is an effective way to model opponnets' goals. Directly using $b_{ij}^{K-1,T_{max}}(g_j)$ as the likelihood to perform Bayesian update may be a bit aggressive, which may lead to poor modeling of opponents adopting mixed strategies. An alternative avenue for exploration is to directly model the probability distribution of the goal based on the initial state of the current episode, i.e., $Pr(g_j | s^{K, 0})$.
>
> For inter-ToM, we consider how history discloses its choice of goals. An effective method is Monte Carlo estimation $\frac{1}{K} \sum_{l = 0}^{K-1} \mathbb{1}(g_j^l = g)$. However, when opponent behavior changes over time, old records will cause estimation bias, so we adopt time-discounted estimation$\frac{1 - \alpha}{1 - \alpha^K}\sum_{l = 0}^{K-1} \alpha^{K-1-l}\mathbb{1}(g_j^l = g)$ where Equation 2 implies.

---

> ### Author Response · Authors · 2023-11-18
> **Response to Reviewer v2eY (Part 2)**
>
> > Our use of goals
>
> We would like to clarify the process of training with goal. During each episode, PToM will not obtain the opponent's ground-truth goal. After an episode ends, if PToM observes the opponent's complete trajectory, PToM can parse out the goal actually accomplished by the opponent in this episode and put it into the replay buffer; if the opponent's complete trajectory is not observed (as may happen in SSH), the opponent's goal cannot be obtained, and this trajectory will not participate in training. Since the goals in the environment have relatively clear meanings (and correspond to cooperation and defection in social dilemmas), such parsing does not consume many additional resources.
>
> We would also like to note that no matter in the training or evaluation phase, opponents are not required to have explicit goals. All of the learning baselines, LOLA, SI, A3C, and PS-A3C,  do not explicitly model goals. But PToM assumes that the opponent has a goal, or more broadly, mental state, and infers this assumed mental state through opponent's behavior. This is a way to use ToM in agent modeling.
>
> > Belief space planning
>
> The incorporation of belief-space planning is indeed an insightful concept. Beyond addressing goal uncertainty, it also presents an opportunity to handle state uncertainty, thereby extending the applicability of PToM to partially observable environments. Due to the complexity of planning in I-POMDP and our focus on few-shot adaptation in SSDs, we have opted not to use belief-space planning. Nonetheless, we firmly believe that exploring belief-space planning is a crucial future direction. Such an approach could potentially extend the applicability of this algorithm to more realistic and a broader range of environments, including those beyond SSD settings. We also advocate for increased attention from the multi-agent community towards the synergy between ToM and planning. This combination holds significant promise as an effective methodology in dealing with social interactions, particularly those requiring few-shot decision-making.
>
>
>
> > Markov Games
>
> Thank you! Markov Game is indeed more appropriate. We have revised the manuscript.
>
>
>
> > Boltzmann model
>
> Yes, this is mainly to encourage exploration while training. During evaluation, we will use argmax, that is, $\beta \to +\infty$ in the Boltzmann model. This is also a commonly used setting in the RL community.
>
>
>
> > About word choice
>
> Thanks for your advice! We have replaced "crippled" by "Ablated".
>
>
>
> > Fast convergence of LOLA in SPD
>
> Indeed, LOLA's convergence speed at SPD seems to be fast. However, we observe that there is still a large standard error after LOLA reaches the convergence reward, and the standard error slowly decreases thereafter, which may prove that LOLA' s policy is not stable when it just reaches convergence reward. What's more, after considering the review from Reviewer ciXi and Reviewer zWBc, we have deleted the comparison in convergence speed with the baseline.

---

### Official Review · Reviewer_PVXH · 2023-11-02

**Soundness:** 3 good
**Presentation:** 3 good
**Contribution:** 3 good
**Rating:** 6
**Confidence:** 3

**Summary:**

The paper proposes a multiagent training paradigm for sequential social dilemma (SSD) environments that involves opponent modeling and MCTS planning. The opponent modeling module has two components: goals modeling and goal-conditioned policies. The goal modeling comprises two parts, one being updated through an episode in a Bayesian manner (intra-ToM) and another (inter-ToM) forming a prior. The goal-conditioned policies are learned using a replay buffer and a cross-entropy loss. The planning is performed for each agent separately by sampling the opponents' objectives from the belief distribution supplied by the goals modeling part and using the appropriate goal-conditioned policies. This procedure is repeated multiple times, and the averaged Q-values induce the agents' Boltzmann policies. The method was evaluated on three SSDs.

**Strengths:**

* The paper proposes a method that successfully blends multiagent setup, planning, and opponent modeling.
* The results are promising and show the adaptation capabilities of the approach.

**Weaknesses:**

* The paper deals with toy environments. What are the scaling capabilities (and limitations) of the approach to more complex tasks, multiple agents, number of rounds, etc.?
* The method compares against other baselines (including model-free approaches), however, it is not clear how fair this comparison is. One has to take into account multiple factors: planning budget, number of repeated planning runs, time to train each of the networks, number of parameters, number of network inferences, wall time, etc.
* The paragraph below Table 1 is unclear.
* The last paragraph of Section 5.2 SSH seems rather anecdotic. Is there evidence that what is described happens (visualization of beliefs, mathematical argument, etc.)?
* In Section 5.2 SSD, the paper mentions "the effectiveness of PToM in quickly adapting to non-cooperative behavior". A justification or analysis of this phenomenon would be helpful (visualization of beliefs, mathematical argument, etc.). Additionally, some insights into the lack of adaptivity of the baseline models would improve the exposition.
* In Section 5.2 SPD, it is written that "PToM tends to engage in exploratory cooperative actions to understand the opponent’s characteristics". What is the reasoning behind this statement?

Other:
* The overall mechanics of MCTS (expansion phase, tree-traversal, backpropagation step, action-taking) could be explained better.
* The paper would benefit from moving the pseudo code from Appendix A to the main paper.
* The parameters used for each environment should be placed in the main body of the paper (in particular, the number of seeds). Consequently, the Authors should consider moving Table 3 from Appendix D.3 to the main paper.
* The content of Appendix F (emergence of social intelligence) is interesting enough that the Authors should consider moving it to the main body.

**Questions:**

See above.

---

> ### Author Response · Authors · 2023-11-18
> **Response to Reviewer PVXH (Part 1)**
>
> > Scaling capabilities
>
> Thanks for you comments! In this manuscript, we test the performance of our approach in the systems of 4 agents. Our approach is decentralized-training-decentralized-execution, not restricted by the number of agents and the number of rounds. It can be readily applied to the sequential decision-making problems among multiple agents and various number of rounds. The environments we used aims to describe the nature of social dilemmas where independent agents need to decide when to compete against or to cooperate with others, to recognize who is the friend and who is the foe, and to plan how to attract the targeted agent to achieve cooperation. These environments are symbolic and look toy, but describe complex interaction pattern. Of course, in the future work, we will do test our approach in more realistic environments where the ability of perception and control are tested as well. This is the way to evaluate the practicability of our approach.
>
>
>
> > Comparison in sample efficiency
>
> Thank you very much for your advice! We agree with you that Figure 3 does not show a reasonable comparison of sample efficiency between our approach and the model-free baselines. We have removed Figure 3 and corresponding content from the manuscript. In addition, to compare the performance between our approach and other planning algorithms, we have used the direct-OM ablation, which can be considered as the implementation of vanilla MCTS,  as a planning baseline. Please find corresponding results in Table 1 and 2 in the revised manuscript.
>
>
>
> > The ambiguity of the paragraph
>
> Thanks for your comment! We think this ambiguity comes from the introduce of LI-Ref and improper metrics of algorithms' performance. In the manuscript, algorithms' performance is measured by the average reward over a specific period of timestep, which is not very informative. Although we introduced LI-Ref (a well-established RL agent performing adaptation to other agents through extensive interactions) as a reference to compare and evaluate algorithms' few-shot adaptation ability, it is placed in Appendix F separately, not well-integrated into Table 1. Inspired by your comments, we have adjusted the metric to be the normalized score. Corresponding results can be found in Table 1 (self-play performance) and Table 2 (few-shot adaptation performance) in the revised manuscript. With regard to the paragraph below Table 1, it is removed from the revised manuscript.  The detailed description of LI-Ref is given in Appendix F.1.
>
>
>
> > Belief visualization
>
> Thanks very much for your advice! We have added the visualization of belief updates in the adaptation phase. please find the figure and corresponding analysis in Appendix F.2 in the revised manuscript. This visualization substantiates our experimental claims. Thanks again for your advice!
>
>
>
> >  Exploratory cooperative actions of PToM in SPD
>
> When observing agent behavior, we see PToM has shown exploratory cooperative actions. The figures of trajectories can be found https://anonymous.4open.science/r/PToM4SSD-F068 . (Currently, the figures are not exquisite enough. We are sorry for that. We will polish these figures and add them into the revised manuscript later. We hope these rough figures can accurately express PToM's trajectory in adaptation to unseen opponents.) In each figure, the green line with arrows represents the trajectory of PToM at the beginning of an episode, with arrows pointing in the direction of movement.  The light blue and light red regions represent the river where bags of waste may appear and the forest where apples may grow, respectively. In Figure 1 PToM is facing three cooperators, while Figure 2 shows the situation of facing three exploiters. Figures show that PToM may move towards the rubbish area, indicating attempts of cooperative behavior. These attempts will be very important when interacting with conditional cooperators who will cooperate if there are already some cooperators in the environment. This type of trajectory is not found in LOLA, SI and A3C. When facing three cooperators, the average number of steps for LOLA, SI, and A3C in the rubbish area are 1.3, 0.5, and 3.1 respectively, while the average number of steps not in the apple area are 6.2, 4.8, and 11.0. The corresponding quantities for PToM are 14.1 and 22.8. This also reflects a cooperative tendency of PToM. When facing three exploiters, each algorithm spends a large amount of time away from the apple area, and only PToM shows an obvious trajectory to the rubbish area as mentioned above, while the movements of other algorithms look random.

---

> ### Author Response · Authors · 2023-11-18
> **Response to Reviewer PVXH (Part 2)**
>
> > Mechanics of MCTS
>
> As the space limitation, the main paper introduces MCTS briefly. We have added two reference papers [Silver & Veness, 2010; Liu et al., 2020] which give a detailed and clear description of MCTS.
>
>
>
> > Moving some contents from appendix to the main paper
>
> Thanks very much for your comments! Due to space limitation, we have to put the mentioned contents in the Appendix. In the revised manuscript, we further clarify the references to the Appendix in the main text. We hope this works.

---

### Official Review · Reviewer_zWBc · 2023-11-06

**Soundness:** 2 fair
**Presentation:** 3 good
**Contribution:** 2 fair
**Rating:** 3
**Confidence:** 4

**Summary:**

This article proposes an algorithm for multi-agent systems that incorporates the benefits of planning and of opponent modelling. The algorithm is validated on custom environments that are temporally and spatially extended versions of the three main social dilemma incentive structures. The algorithm is constrasted against LOLA, Social Influence, traditional A3C and prosocial A3C and shows much faster and better convergence than the baselines. The authors also show their algorithm is much better a few-shot adaptation to new co-players.

**Strengths:**

The article is well written and for the most part explains the methodology in enough detail. The combination of Theory of Mind (ToM) and Monte-Carlo Tree Search (MCTS) is interesting, and valuable. The authors sidestep the biggest hurdle with applying MCTS to multi-agent situations by training a module that predicts actions conditioned on (hidden) co-player goals. When doing MCTS, the search samples goals for co-players, and uses the learned conditioned policies to do the rollouts. The algorithm is, prima facie, significantly more sample efficient than the studied baselines. The presentation of results on the three custom environments is appreciated, as it shows behaviours on the main social dilemma categories.

**Weaknesses:**

The fundamental problem with this work is that it is not comparing fairly against baselines. No planning benchmark was given, despite talking about MuZero or even vanilla MCTS. I understand that opponent modelling restricts applications of straight-forward MCTS, but even if one were to assume random opponents, or full self-play (i.e. using the self policy for rollouts of others) it would produce a reference point. Otherwise, comparing how many environment steps it takes to converge is unfair for something that does search with something that doesn't.

The question of few-shot adaptation is also suspect. The main issue is that their algorithm is specifically designed to take an inter- and intra-episode estimation of co-player goals. None of the baselines have this capability. Then the authors test how fast their agent can adapt to others in a fixed number of environment timesteps at inference time. This is unfair for non-MCTS, non-goal conditioned algorithms which are _not_ built for this purpose, and couldn't even process the goals signals. Moreover, that are specific benchmarks for few and zero-shot generalisation to others (e.g. Leibo, et al 2021, cited). Those seem to be too complex for this algorithm to tackle, though.

Another significant issue is that it is unclear whether the environments are partially observable. I suspect not, since MCTS would be much harder to roll out if so. However, the authors do talk about the POMDP (Partially Observable Markov Decision Process) as the conceptualisation typically used in studying sequential social dilemmas. If this algorithm _only_ works on fully observable environments, that severely limits the applicability. I'd like to see a scalable solution that does apply to POMDPs to warrant publication in ICLR.

Even beyond the issue of full observability, the use of custom environments is a drawback. I'm not convinced that the environments actually have the incentive structure the authors claim. When creating a temporally and spatially extended version of a repeated matrix game, one has to be careful that the strategies actually map to the right returns. The traditional way to validate this is via a Schelling Diagram (as explained in Hughes, et al 2018, cited). The authors sequential stag hunt wouldn't have the right incentives if the episode were allowed to continue past the first hunt. But this causes those episodes to be artificially short. This, when paired with a system in which evaluation is a fixed number of timesteps, for environments that have very different lengths of episodes, makes me wonder why such a system was chosen. I worry it might be to artificially strengthen the results (as in, cherry picked numbers).

**Questions:**

What do the authors mean when they say: "We observe the emergence of social intelligence"? It seems to me this is either a strong claim (i.e. none of the other algorithms exhibit it) for which I would need a definition, or a weak claim, in which case probably all other baselines also exhibit it.

How is the PToM agent learning the goals? Is it trained against specifically incentivised co-players? Otherwise how does the ground-truth goal is obtained?

---

> ### Author Response · Authors · 2023-11-18
> **Response to Reviewer zWBc (Part 1)**
>
> > Comparison in sample efficiency
>
> Thank you very much for your valuable advice! We agree with you that Figure 3 does not show a reasonable comparison of sample efficiency between our approach and the model-free baselines. We have removed Figure 3 and corresponding content from the manscript. In addition, to compare the performance between our approach and other planning algorithms, we have added direct-OM as a baseline. direct-OM can be considered as the implementation of vanilla MCTS.  Please find and the corresponding results in Table 1 and Table 2 in the revised manuscript.
>
>
>
> > Few-shot adaptation
>
> Humans have the ability to properly deal with unfamiliar others within very limited interactions. To interact and cooperate with humans, a truly intelligent agent should also have this ability. More specifically, to cope with this problem, agents need to fast recognize and adapt their behavior to previous unseen others, who may also simultaneously adjust their behaviors to adapt to the agents. This is the few-shot adaptation problem this manuscript focuses on. To address this problem, we propose an approach which hierarchically models other agents' policies (i.e., goal + goal-conditioned policy) and uses ToM to infer other agents' goals and behavior.  However, we would like to clarify that this is only one of possible methods and does not mean that explicit modeling of the goal is necessary to achieve few-shot adaptation. The performance of various methods in few-shot adaptation needs to be explored. The baselines that we compare with all report superior performance in some social dilemmas in the literature we have cited, and some of them also have excellent performance in zero-shot adaptation to others. The few-shot setting is similar to finetune for pretrained models. The difference between it and zero-shot setting is mainly in the adaptation stage. As mentioned above, in the setting of few-shot adaptation, both focal agents and opponents can update the policy, which is not allowed for zero-shot adaptation. All agents are adapting to each other and changing their behavior dynamically, which provides the possibility to improve utility and also brings challenges to training.
>
>
>
> > Partial observability
>
> The environments we focus on are fully observable.In this paper, we concentrate on few-shot adaptation to unseen coplayers in SSDs. To cope with this problem, agents should learn  the environments (i.e. game rules of SSDs), and have the ability to recognize coplayers' behavior pattern from just few actions and the capacity to respond various behavior appropriately. Thus, as the first attempt to study this problem, we ignore partially observability. Undoubtedly, in the future, we will extend our algorithm to partially observable sequential social dilemmas. As the Reviewer v2eY suggested, intergating belief-space planning with PToM will be a promising way to extend the applicability of our approach to partially observable environments.
>
>
>
> > Environment
>
> Thanks for your valuable comments! We have added the Schelling Diagrams for the three environments we used to evaluate the algorithms (please find the Schelling Diagrams in Appendix C in the revised manuscript).
>
> The Schelling diagrams demonstrate that our environments are appropriate extentions of classic matrix-form social dilemmas.
>
> Our temporally and spatially extended environments maintain the nature of the three representative paradigms of social dilemmas.
>
> Please find a detailed analysis of the three social dilemmas and Schelling Diagrams of our environments in Appendix C in the revised manuscript.
>
> We would like to thank the reviewer for the valuable comments again! It helps to demonstrate the validity of our environments.

---

> ### Author Response · Authors · 2023-11-18
> **Response to Reviewer zWBc (Part 2)**
>
> > About social intelligence
>
> Thanks for your question! During analyzing PToM agents' behavior, we incidentally find some interesting phenomena. In the absence of communication, decentralized agents behave collectively (i.e. self-organized cooperation). The agents equipped with Bayesian ToM, which seems to be much simpler than human cognition, try to make an alliance with one coplayer to resist the exploitation by another coplayer. These intelligent behavior emerge in a spontaneous way, without any external guidance or control. What we want to express is that simple and interpretable cognitive model has the potential to achieve complex intelligent behavior. A truly intelligent agent should be able to understand and interact with other agents and human in a human-like way. It is an enduring challenge to recognize, adapt to and understand the underlying mechanisms of social intelligence. We will keep focus on this topic and dig deeper.
>
>
> > Our use of goals
>
> We would like to clarify the process of training with goal. During each episode, PToM will not obtain the opponent's ground-truth goal. After an episode ends, if PToM observes the opponent's complete trajectory, PToM can parse out the goal actually accomplished by the opponent in this episode and put it into the replay buffer; if the opponent's complete trajectory is not observed (as may happen in SSH), the opponent's goal cannot be obtained, and this trajectory will not participate in training. Since the goals in the environment have relatively clear meanings (and correspond to cooperation and defection in social dilemmas), such parsing does not consume many additional resources.
>
> We would also like to note that no matter in the training or evaluation phase, opponents are not required to have explicit goals.  But PToM assumes that the opponent has a goal, or more broadly, mental state, and infers this assumed mental state through the opponent's behavior. This is a way to use ToM in agent modeling. Therefore, we do not need specifically incentivised co-players in training.

---

### Official Review · Reviewer_ciXi · 2023-11-10

**Soundness:** 1 poor
**Presentation:** 3 good
**Contribution:** 2 fair
**Rating:** 3
**Confidence:** 4

**Summary:**

The authors introduce a new method for decision-time planning, based on inferring co-player goals, learning co-player goal-based policies and then rolling out MCTS using these policies. They apply this new method to several sequential social dilemma domains.

**Strengths:**

- The introduction provides clear motivation, and the conclusion provides a concise summary and sound suggestions for future work.
- To my knowledge, the idea of doing MCTS based on explicitly learned co-player models using goals that are inferred online is novel. In principle, this could yield an impactful improvement over the state of the art.
- The method is generally well-described, and therefore I assess that this paper is likely reproducible.

**Weaknesses:**

- The main weakness of this paper is that the experimental results are quite weak. Table 1 does not show a very large improvement from PToM over the baselines except in some specific cases (e.g. adaptation to exploiters in SS). PToM does not achieve cooperation in SPD. There are no videos or behavioral analyses provided to corroborate qualitative claims about the better behaviour of PToM over the baselines. Can the authors provide such results? For instance, what is the evidence that PToM is significantly better at hunting stags in SSH?
- Some of the experimental claims are poorly explained and some of them are unsubstantiated. For example, on page 8, what is meant by "we find that the leading space of PToM is not significant"? Also on page 8, the "further intuition" on why PToM is effective at adaptation is not substantiated by analysis of the belief model for the agent during intra- and inter-ToM. Can the authors provide this data?
- There is some important missing related work which should be cited on theory of mind in the context of RL: https://arxiv.org/pdf/2102.02274.pdf, https://arxiv.org/pdf/1901.09207.pdf. Ideally these methods would be provided as baselines, or the authors would explain why their method was clearly an improvement from a theoretical standpoint. Can the authors comment on this?
- The authors could also cite more recent work in the LOLA line: e.g. https://arxiv.org/abs/2205.01447.
- Above equation (3) the authors seem to assume that the focal agent has access to the goals of its opponents during training. Is this correct? Yet earlier in Section 3, the authors claim that agent j's true goal is inaccessible to agent i. Can they clarify this apparent contradiction (perhaps the goals are known during training but not during execution, in the usual centralized training, decentralized execution paradigm)?
- In Figure 3, does the x-axis for PToM take into account the experience steps used in MCTS? This is unclear, and if not, these experience steps should also be accounted for, to make this a fair comparison between algorithms.

**Questions:**

See Weaknesses.

---

> ### Author Response · Authors · 2023-11-18
> **Response to Reviewer ciXi (Part 1)**
>
> > Experimental results and analysis
>
> Thanks for your helpful comment! We believe the seemingly week results are caused by our improper metric. In the manuscript, algorithms' performance is measured by the average reward over a specific period of timestep, which is not very informative. Although we introduced LI-Ref (a well-established RL agent performing adaptation to other agents through extensive interactions) as a reference to compare and evaluate algorithms' few-shot adaptation ability, it is placed in Appendix E separately, not well-integrated into Table 1. Inspired by your comments, we have adjusted the metric to be the normalized score. Corresponding results can be found in Table 1 (self-play performance) and Table 2 (few-shot adaptation performance) in the revised manuscript. The results show that the reward achieved by PToM and LOLA is already close to the maximum possible reward, thus the advantage of PToM is not significant compared to LOLA. (This is also what the sentence "we find that the leading space of PToM is not significant" tries to express. This sentence has been deleted in the revised manuscript.) In SS and SPD, PToM outperforms other algorithms in 6 out of 7 and 4 out of 7 adaptation scenarios, respectively. After taking variance into consideration, other algorithms achieve best results in no more than 3 adaptation scenarios for each environment. These consistent advantages across different environments demonstrate the effectiveness and robustness of PToM.
>
> Regarding the failure of achieving cooperation in SPD, PToM is designed to be self-interested (i.e. rational), aiming to maximize own reward, regardless of others' benefits. Due to the inherent complexity of SPD, self-interested agents, PToM as well as  LOLA, SI and A3C, tend to select the short-term optimal strategy and fall into the dilemma of total defection. In SPD, only the agents who are long-sighted and take into account others' reward can escape from the dilemma. It is our future work that realizing such agent through developing decentralized-training-decentralized-execution algorithms. It is a great challenge for DTDE algorithms,  and we consider it to be an important future work.  Furthermore, what needs to be emphasized is that our main goal in evaluating PToM and the baselines is not to achieve comprehensive cooperation, but rather to assess their ability to adapt quickly to previously unseen opponents and make appropriate response decisions. In particular, when facing self-interested opponents in SPD, we expect that the focal agent does not cooperate and avoids being exploited by them.
>
> Thanks for your helpful comments on the explanation of experimental results. In the revised manuscript, we have added the visualization of belief updates and corresponding analysis (see details in Figure 3 in Appendix F.2).
>
> We also visualize PToM's behavior trajectories in adaptation to unseen opponents. The figures of trajectories can be found https://anonymous.4open.science/r/PToM4SSD-F068. (Currently, the figures are not exquisite enough. We are sorry for that. We will polish these figures and add them into the revised manuscript later. We hope these rough figures can accurately express PToM's trajectory in adaptation to unseen opponents.). In each figure, the green line with arrows represents the trajectory of PToM at the beginning of an episode, with arrows pointing in the direction of movement.  The light blue and light red regions represent the river where bags of waste may appear and the forest where apples may grow, respectively. In Figure 1 PToM is facing three cooperators, while Figure 2 shows the situation of facing three exploiters. Figures show that PToM may move towards the rubbish area, indicating attempts of cooperative behavior. These attempts will be very important when interacting with conditional cooperators who will cooperate if there are already some cooperators in the environment. This type of trajectory is not found in LOLA, SI and A3C. When facing three cooperators, the average number of steps for LOLA, SI, and A3C in the rubbish area are 1.3, 0.5, and 3.1 respectively, while the average number of steps not in the apple area are 6.2, 4.8, and 11.0. The corresponding quantities for PToM are 14.1 and 22.8. This also reflects a cooperative tendency of PToM. When facing three exploiters, each algorithm spends a large amount of time away from the apple area, and only PToM shows an obvious trajectory to the garbage area as mentioned above, while the movements of other algorithms look random.

---

> ### Author Response · Authors · 2023-11-18
> **Response to Reviewer ciXi (Part 2)**
>
> >Baselines and theoretical analysis
>
> Thank you for providing the related work! We have cited these papers in our revised manuscript. The paper https://arxiv.org/pdf/2102.02274.pdf  provides a recursive deep generative model to learn belief states of other agents at different orders by sampling latent variables. The framework PR2 proposed by  https://arxiv.org/pdf/1901.09207.pdf explicitly takes into account the influence of my action on opponent policy when inferring opponent policy. Due to the time constraint, we cannot give experimental results of PR2-AC currently. Similarly, when modeling other agents, the existing baseline LOLA considers the impact of my policy on the policy gradients of other agents, and SI considers the influence of my action on other agents' policies. What's more, LOLA and SI have been tested in social dilemmas and performed well. Thus, considering our focus on sequential social dilemmas, we think LOLA and SI are representatives of MARL algorithms equipped with the ability of recursively modeling others.
>
> We have tried to give a theoretical analysis of our approach in SSDs. However, as the complexity and non-stability of the environments, we failed to give theoretical guarantees. Instead, we give a concise theoretical analysis of our approach in two-player matrix-form games (please see details in Appendix B). This analysis demonstrates the convergence of PToM to the optimal response strategy. Meanwhile, empirical results show the consistent improvement of PToM, although the advantage of PToM is not very significant in some cases. Ablation study indicates that hierarchical modeling of other agents and ToM based inference on others' goals are crucial to the adaptation to unseen opponents.
>
>
> > LOLA citation update
>
> Thank you for providing the related work! We have added this paper in the revised manuscript.
>
>
>
> > Our use of goals
>
> We would like to clarify the process of training with goal. During each episode, PToM will not obtain the opponent's ground-truth goal. After an episode ends, if PToM observes the opponent's complete trajectory, PToM can parse out the goal actually accomplished by the opponent in this episode and put it into the replay buffer; if the opponent's complete trajectory is not observed (as may happen in SSH), the opponent's goal cannot be obtained, and this trajectory will not participate in training. Since the goals in the environment have relatively clear meanings (corresponding to cooperation and defection in social dilemmas), such parsing does not consume many additional resources.
>
> We would also like to note that no matter in the training or evaluation phase, opponents are not required to have explicit goals. All of the learning baselines, LOLA, SI, A3C, and PS-A3C,  do not explicitly model goals. But PToM assumes that the opponent has a goal, or more broadly, mental state, and infers this assumed mental state through the opponent's behavior. This is a way to use ToM in agent modeling. Therefore, we do not consider the training to be centralized, since we do not get the ground-truth goal information or opponents' models through a centralized module, and even the opponent algorithm does not have a goal variable as ground-truth.
>
>
>
> > Comparison in sample efficiency
>
> Thank you very much for your advice! We agree with you that Figure 3 does not show a reasonable comparison of sample efficiency between our approach and the model-free baselines. We have removed Figure 3 and corresponding content from the manuscript. In addition, to compare the performance between our approach and other planning algorithms, we have added direct-OM as a baseline. Direct-OM can be considered as the implementation of vanilla MCTS.  Please find and the corresponding results in Table 1 and Table 2 in the revised manuscript.

---

> ### Comment · Reviewer_ciXi · 2023-11-22
>
> I thank the authors for their response. In reply:
>
> **Experimental results and analysis**
>
> In my view the decision to present the results as normalized scores makes the paper *weaker*, not stronger. This feels to me like the authors are massaging the data presentation to fit their narrative. As far as I can tell, their method does not succeed in significantly improving the state of the art performance on sequential social dilemmas, even where there is quite a large amount of headroom to improve on existing algorithms.
>
> The authors would do better to return to the drawing board for their algorithm and identify modifications to the algorithm which will provide strong improvements in individual and collective return over existing algorithms.
>
> The provided trajectory plots (https://anonymous.4open.science/r/PToM4SSD-F068) are not at all informative enough. In order to accurately and reliably assess the behaviors, one would need to see videos from multiple episodes of the behaviour of all agents, and to compare the qualitative and quantitative differences in individual and group-level behaviours between different algorithms. I recommend that the authors make this a standard part of their workflow should they pursue this line of research.
>
> **Other comments**
>
> The provision of the Direct-OM baseline is a welcome addition. However, my remaining concern about the strength of the empirical results, coupled with the lack of precise argument / empirical results about how this improves over the existing work in this area (https://arxiv.org/pdf/2102.02274.pdf, https://arxiv.org/pdf/1901.09207.pdf) means that I cannot recommend this for acceptance in its current form.

---

> ### Author Response · Authors · 2023-11-23
> **Thank you for your valuable feedback**
>
> We sincerely appreciate your valuable feedback and thank you for dedicating your time and effort.
>
> While it may appear that the improvements achieved by PToM are not substantial, it is important to emphasize the consistent enhancements it demonstrates when compared to state-of-the-art algorithms. This consistency highlights the advantages of our algorithm. In addition, PToM, along with the structure of "ToM+planning" it represents, has novelty and great potential in adaptation to unseen opponents, which underscores the valuable contributions our manuscript offers to the community.
>
> Regarding the display of normalization scores, our intention is to adopt a standardized presentation method for results across various environments and scenarios. This approach aims to enhance the readability of our article, aligning with common practices observed in previous research (Sunehag et al., 2017; Leibo et al., 2021, cited). We think that the normalization process itself does not make the results stronger or weaker. Comparing the results of the original rewards and that of the normalized scores, you can find that there is no qualitative difference between these two metrics. They both demonstrate our conclusion that PToM consistently surpasses baselines.
>
> These trajectory plots are given to illustrate the behavior of PToM in few-shot adaptation to unseen opponents. Beside the trajectory plots, showing PToM’s exploratory cooperation behavior, we do observe the behavior of all agents from multiple episodes, and make a comparison between different algorithms. A statistical analysis of these behavior is given as “When facing three cooperators, the average number of steps for LOLA, SI, and A3C in the rubbish area are 1.3, 0.5, and 3.1 respectively, while the average number of steps not in the apple area are 6.2, 4.8, and 11.0. The corresponding quantities for PToM are 14.1 and 22.8. This also reflects a cooperative tendency of PToM. When facing three exploiters, each algorithm spends a large amount of time away from the apple area, and only PToM shows an obvious trajectory to the garbage area as mentioned above, while the movements of other algorithms look random.” Such exploratory cooperative behavior is effective when adapted to conditional cooperators, which are the types of some dominant strategies in prisoner's dilemmas such as TFT.
>
> In addressing the selection of baselines, we fully recognize the importance of the algorithms that you mentioned. However, it is essential to note that our research primarily focuses on social dilemmas. Unlike the baselines presented in our current article, the works you mentioned (https://arxiv.org/pdf/2102.02274.pdf, https://arxiv.org/pdf/1901.09207.pdf) do not report their performance specifically in social dilemma environments. Consequently, making comparisons with these works may not fully showcase our improvements relative to the existing state-of-the-art. We acknowledge the potential value of including comparisons with these algorithms and are actively working towards testing these baselines. Nevertheless, we believe that the current set of baselines offers sufficient context for our findings.
>
> Once again, we express our gratitude for your comments. We hope that this explanation provides a clearer perspective on the value of our article, and we remain committed to enhancing its quality.

---

### Author Response · Authors · 2023-11-18
**Major revision in our revised manuscript**

In our revised manuscript, several key enhancements and refinements have been made to the content:

- We used direct-OM as a model-based baseline, which can be considered as an implementation of vanilla MCTS.
- We removed comparison in sample efficiency between our approach and the model-free baselines.
- We implemented reward normalization with normalization bounds set by the rewards of LI-Ref and the random policy.
- We added Schelling diagrams of our environments in Appendix C.
- We added visualization of belief updates in Appendix F.2.
- We added some citations to enrich the manuscipt with a more extensive information base.

---

### Meta-Review · Area_Chair_NsfH · 2023-12-06

**Metareview:**

The algorithm does not cooperate in sequential social dilemmas, but the exposition of the paper wrongly claims it to do so. The authors said so themselves in their response that there's no particular reason to expect it to cooperate since it's a self-interested model. So it seems that the actual construction of the algorithm is at odds with the structure of the paper, as well as the choice of enviroments for validation.

One of the biggest problems is that when reviewers mentioned the poor results the authors responded by changing the paper to use a normalized statistic that obscures the poor results (which they attempted to justify by saying the Melting Pot paper used such a normalized statistic, but in that case there was a special "exploiter" agent in the normalization set which trained on the test set to get an artificially high score, this was specifically so the normalized scores would still be meaningful in the case of unsolved problems where all the contestants got low scores.) Here introducing the normalization without the exploiter agent only serves to obscure the fact that the focal agent is not cooperating.

Anyway, I did appreciate the Schelling diagrams. Thank you for adding them.

I would urge the authors to take the advice of reviewer ciXi and the others. I do think that this paper is close to being accepted, and with a bit more work it should get in.

**Justification For Why Not Higher Score:**

See above

**Justification For Why Not Lower Score:**

N/A

---

### Decision · Program_Chairs · 2024-01-16

Reject